# A predictive computational platform for optimizing the design of bioartificial pancreas devices

Alexander U. Ernst[1], Long-Hai Wang[1,5] ✉, Scott C. Worland[1], Braulio A. Marfil-Garza[2], Xi Wang[1], Wanjun Liu[1], Alan Chiu[1], Tatsuya Kin[2,3], Doug O'Gorman[2,3], Scott Steinschneider[1], Ashim K. Datta[1], Klearchos K. Papas[4], A. M. James Shapiro[2,3] & Minglin Ma[1] ✉

The delivery of encapsulated islets or stem cell-derived insulin-producing cells (i.e., bioartificial pancreas devices) may achieve a functional cure for type 1 diabetes, but their efficacy is limited by mass transport constraints. Modeling such constraints is thus desirable, but previous efforts invoke simplifications which limit the utility of their insights. Herein, we present a computational platform for investigating the therapeutic capacity of generic and user-programmable bioartificial pancreas devices, which accounts for highly influential stochastic properties including the size distribution and random localization of the cells. We first apply the platform in a study which finds that endogenous islet size distribution variance significantly influences device potency. Then we pursue optimizations, determining ideal device structures and estimates of the curative cell dose. Finally, we propose a new, device-specific islet equivalence conversion table, and develop a surrogate machine learning model, hosted on a web application, to rapidly produce these coefficients for user-defined devices.

The transplantation of pancreatic islets and stem cell-derived insulin-producing cell clusters (SC-βs) into the portal vein of the liver has successfully restored glycemic regulation in patients with type 1 diabetes (T1D); however, the pool of eligible recipients of this procedure is limited, in part, because patients must endure chronic immunosuppression to prevent the immune destruction of the cells[1]. Encapsulating islets or SC-βs in a semipermeable delivery system may protect cells from the immune system while enabling bidirectional mass transfer of nutrients and therapeutics. This approach, constituting a bioartificial pancreas (BAP) device, is thus an attractive strategy for T1D treatment[2,3]. However, while encapsulation prevents immune attack, it also introduces challenges related to mass transfer, especially that of the oxygenation of the encapsulated cells.

Foremost, β-cells, which in their native state are located adjacent to islet-penetrating blood vessels[4], receive oxygen supply in BAP devices only via gradient-driven passive diffusion from the host site[5]. The high oxygen consumption rate (OCR) of β-cells owing to the high metabolic demand of insulin secretion, the relatively low oxygen tension ($pO_2$) in accessible extravascular transplantation sites (Supplementary Table 1), and the low permeability of oxygen in encapsulation devices (usually hydrogel-based) and biological tissue, all further exacerbate oxygen limitations. Acute hypoxia arising from the combination of these factors can lead to cell death, precipitating the release of danger-associated molecular patterns (DAMPs)−molecules that stimulate a more aggressive immune response to the graft[6,7]. It is therefore especially desirable to ensure

[1]Biological and Environmental Engineering, Cornell University, Ithaca, NY, USA. [2]Department of Surgery, University of Alberta, Edmonton, AB, Canada. [3]Clinical Islet Transplant Program, University of Alberta, Edmonton, AB, Canada. [4]Department of Surgery, University of Arizona, Tucson, AZ, USA. [5]Present address: Department of Polymer Science and Engineering, University of Science and Technology of China, Hefei, Anhui, China. ✉ e-mail: hiwang@ustc.edu.cn; mm826@cornell.edu

that oxygen is available to all cells at a level sufficient to sustain their survival. But even if acute anoxia is avoided, insulin secretion—the therapeutic objective—is compromised at moderate $pO_2$ levels[8]. Accordingly, graft function is remarkably sensitive to cellular oxygen availability, which, in turn, is dependent on device design and the physiochemical properties of the encapsulated cells. It is thus of considerable value to develop a quantitative model of oxygen transport and the associated effects on cell survival and insulin secretion in BAP devices to effectively predict their function, as well as to guide and optimize their design.

Quantitative models of mass flows at the scale of the islet cell cluster are well formulated[9–12]; however, existing models at the device scale provide limited insights because they overlook critical stochastic features intrinsic to such devices. Models often consider islets as circles or spheres of uniform diameters[13–17], distributed uniformly[14,17,18], or as part of a composite domain with the rest of the device[16,19] (Supplementary Fig. 1). In practice, isolated mammalian islets are found in a large range of sizes (<50 to >500 μm in diameter) from distributions that vary within and between species[20–23]. Moreover, the cell clusters are nonuniformly distributed in the hydrogel matrix following encapsulation. Therefore, important components of such systems are inherently probabilistic; thus, we hypothesize that this uncertainty exerts considerable influence on the device's expected performance, and further, that computational models that do not capture this are consequently limited and prone to biased predictions.

Herein, we present a computational platform based on the stochastic finite element method[24] we call SHARP ("Simulated Heterogeneity and Randomness Program"), which allows the treatment of model inputs as probabilistic properties. Primarily, islets and SC-β cell clusters are treated as size distributed with diameters sampled from empirically determined probability densities and randomly dispersed in the encapsulating hydrogel, after which Monte Carlo simulations of mass transfer may be performed to determine the expected performance of a user-defined device.

First, we characterize the size distributions of islets and SC-βs from six common sources. Then, as a demonstration of SHARP's capability, we show that the intrinsic uncertainty of human islet size distributions is expected to contribute a substantial impact on the expected insulin secretion capacity of a BAP device, in proportion to the preponderance of larger islets in the isolation. We then validated this prediction in a streptozotocin (STZ)-induced diabetic mouse model using a device containing rat islets with artificially manipulated size distributions. Next, SHARP was leveraged to optimize device design and calculate the expected cell dose required to achieve a functional cure in a standard T1D patient. Finding a strong size dependence of islet outcomes, we furthermore propose adjustments, according to expected functionality, to the islet equivalence conversion table on a device-specific basis, which more accurately reflects their therapeutic potential than the currently used islet mass-based conversion formula. Finally, we developed an ensemble machine learning model to advance the utility of SHARP, which, as a proof-of-concept, rapidly produced the functional adjustments to the islet equivalence conversion table for many BAP devices. In addition to providing insights into modeling inherently stochastic systems, the work herein presents a versatile computational platform to predict the viability of BAP devices and presents general strategies to improve their design.

## Results

### Size distribution characterizations of various islet/SC-β sources

Size distributions of common islet and SC-β sources were first characterized (Fig. 1). Natural cellular replication dynamics result in cell clusters whose diameter ($d$) frequencies are described by either the probability of the lognormal distribution ($f_L$)

$$f_L(d) = \frac{1}{d\alpha\sqrt{2\pi}} \exp\left( \ln \frac{\left(\frac{d}{\beta}\right)^2}{2\alpha^2} \right) \quad (1)$$

or the Weibull distribution ($f_W$)

$$f_W(d) = \frac{\alpha}{\beta} \left(\frac{d}{\beta}\right)^{\alpha-1} \exp\left( -\left(\frac{d}{\beta}\right)^{\alpha} \right) \quad (2)$$

respectively, where $\alpha>0$ is the shape parameter and $\beta>0$ is the scale parameter for each type of distribution (Supplementary Fig. 2)[23]. On this basis, we assumed that the diameters of islets were continuous random variables given by one of these distribution types. Isolations from mice, rats, juvenile pigs, humans, and SC-βs produced from Novo Nordisk (NN SC-βs) and the Professor Jeffrey Millman group at Washington University in St. Louis (WU SC-βs) were imaged and the islet sizes were measured. Because most islet morphologies are not perfectly spherical, we defined the effective diameter, $d_e$, of the individual cell clusters as the diameter of a circle with equivalent area of the measured cell cluster (Fig. 1a and Supplementary Fig. 3). Cumulative density functions of the two-parameter lognormal and Weibull distributions (Supplementary Eqs. (S3) and (S4)) were then fit to the measured actual cumulative probabilities on a number- and volume-weighted basis and the one yielding the best fit was determined.

The effective diameter distributions of all primary islet sources were well described by either of these distribution types (Fig. 1b–g, Supplementary Fig. 4a–e and Supplementary Tables 2 and 3). The number-average effective diameter ($\bar{d}_n$) of all primary islet sources was between 120 and 180 μm, while the volume-average effective diameter ($\bar{d}_v$) was between 180 and 220 μm (Supplementary Table 4). Because volume scales cubically with the effective diameter, the distributions, when weighted by volume, were all right shifted relative to those determined on a number basis. Consequently, the low frequency of larger islets in all distributions nonetheless contributed an outsized significance to the proportion of islet volume (e.g., only ~6% of human islets were larger than 200 μm in effective diameter, but this represented ~33% of the total volume). Rat, juvenile porcine, and human islet distributions all exhibited heavy right tails, whereas mouse islets were slightly left-skewed with $d_e \geq 300$ μm comprising a negligible fraction. Furthermore, we note that human islet size distributions were variable depending on their site of isolation (Supplementary Fig. 5), indicating that the size distributions are influenced by the isolation protocol in addition to endogenous differences between pancreases. Expectedly[25], distributions of both SC-β sources were narrower and more symmetrical, but were nonetheless robustly described by Weibull distributions (Supplementary Fig. 4f, g), which was favored over the normal distribution due to the absence of positive probabilities for selecting a diameter with a negative value.

A separate characterization of mouse islets reported a positive correlation between islet size and ellipticity[23]. Our analyses corroborated this finding in mouse islets, to a lesser extent in juvenile porcine islets, and to an even lesser extent in human islets, a trend that interestingly correlated with that of their reported OCRs (Supplementary Table 5 and Supplementary Fig. 6). Increased ellipticity should bestow favorable islet oxygenation, thus we speculate that this phenomenon may indicate a morphological adaptation to hypoxia, hence the greater size dependence of ellipticity in islet sources with high OCRs. Islet ellipticity likely also depends on other factors which were not deliberately controlled here, thus more extensive consideration of this hypothesis may be a subject of a future study.

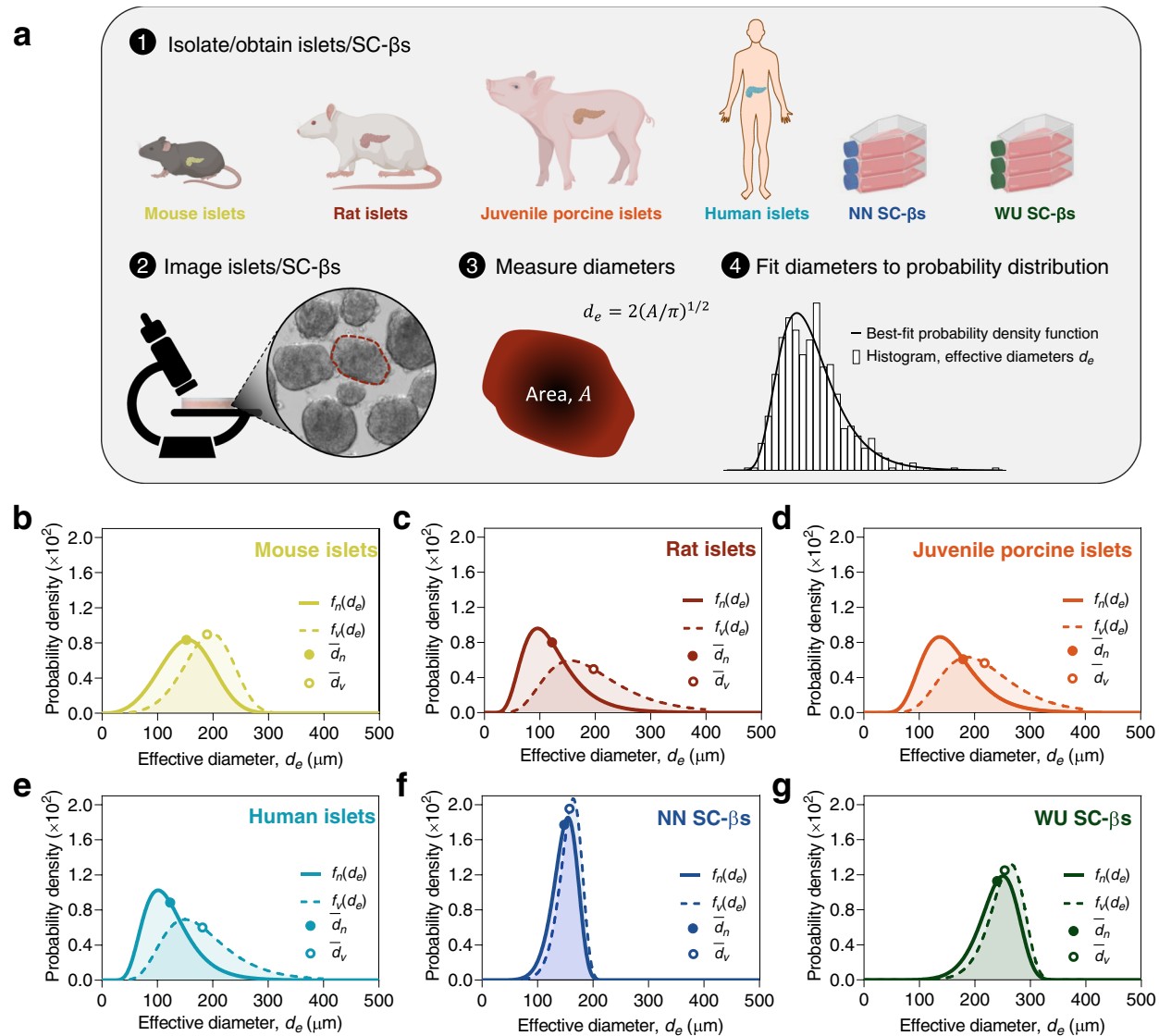

**Fig. 1 | Islet and SC-β size distributions. a** Schematic illustrating the procedure followed for determining effective diameter ($d_e$) distributions of islets from common sources. **b–g** Best-fit probability density functions of $d_e$ of islets from mice, rats, juvenile pigs, and humans, and SC-βs from two sources (NN and WU). Distributions are shown on a number basis ($f_n$, solid lines) and a volume basis ($f_v$, dashed lines). Solid circles indicate the mean effective diameter, $\bar{d}_n$, and open circles indicate the mean effective diameter when weighted by volume, $\bar{d}_v$. Curves represent distributions of aggregated data from multiple batches (Supplementary Tables 2 and 3). Source data are provided as a Source Data file.

## SHARP enables the evaluation of a diverse range of bioartificial pancreas devices

SHARP was then developed to incorporate information about the islet size distributions, OCRs, and random spatial localization to probabilistically analyze the expected performance of BAP devices (Fig. 2). In brief, the program receives cell source, volumetric cell density within the device, and a description of the device's geometry, generates the device in silico treating islets or SC-βs as randomly distributed spheres with diameters sampled from their corresponding size distributions (Supplementary Fig. 7), solves for internal oxygen gradients and related outcomes, and reiterates following the Monte Carlo approach to obtain stochastic results (Fig. 2a). Herein, we consider planar, cylindrical, and hollow cylindrical designs, the latter of which describes a device with a concentric cell-containing hydrogel coating with an acellular passive core (Fig. 2b). These geometries describe or approximate many existing devices, though others, including simple structures such as spherical or toroidal microcapsules[19], or non-parameterizable complex devices, may be examined as well

(Supplementary Fig. 8). Furthermore, the random cell cluster seeding algorithm allowed cell cluster densities over 40% of the total graft volume, which is well above levels conventionally employed (Fig. 2c).

The physical problem is described comprehensively in the Methods (and Supplementary Fig. 9). In brief, oxygen from the host site, assumed to be at a uniform 40 mmHg at the device-host boundary (Supplementary Fig. 9a), is transported in the hydrogel by passive diffusion. Herein, we are concerned with modeling transport in generic (i.e., representative) devices, thus it was necessary to select one material. Because of its widespread use both presently and historically[26], we selected 2% (w/v) alginate as this model material, with the relevant permeability parameters obtained from the literature. In the islets, oxygen transport is governed by the balance between diffusion and consumption, the latter described by Michaelis–Menten kinetics (Supplementary Fig. 9b) with the maximum rate given by the cell source's reported OCR. Three oxygen-related outcomes are calculated for the islet population and for each cell cluster: the volume-average pO₂, the expected fractional volume of necrotic tissue, and the

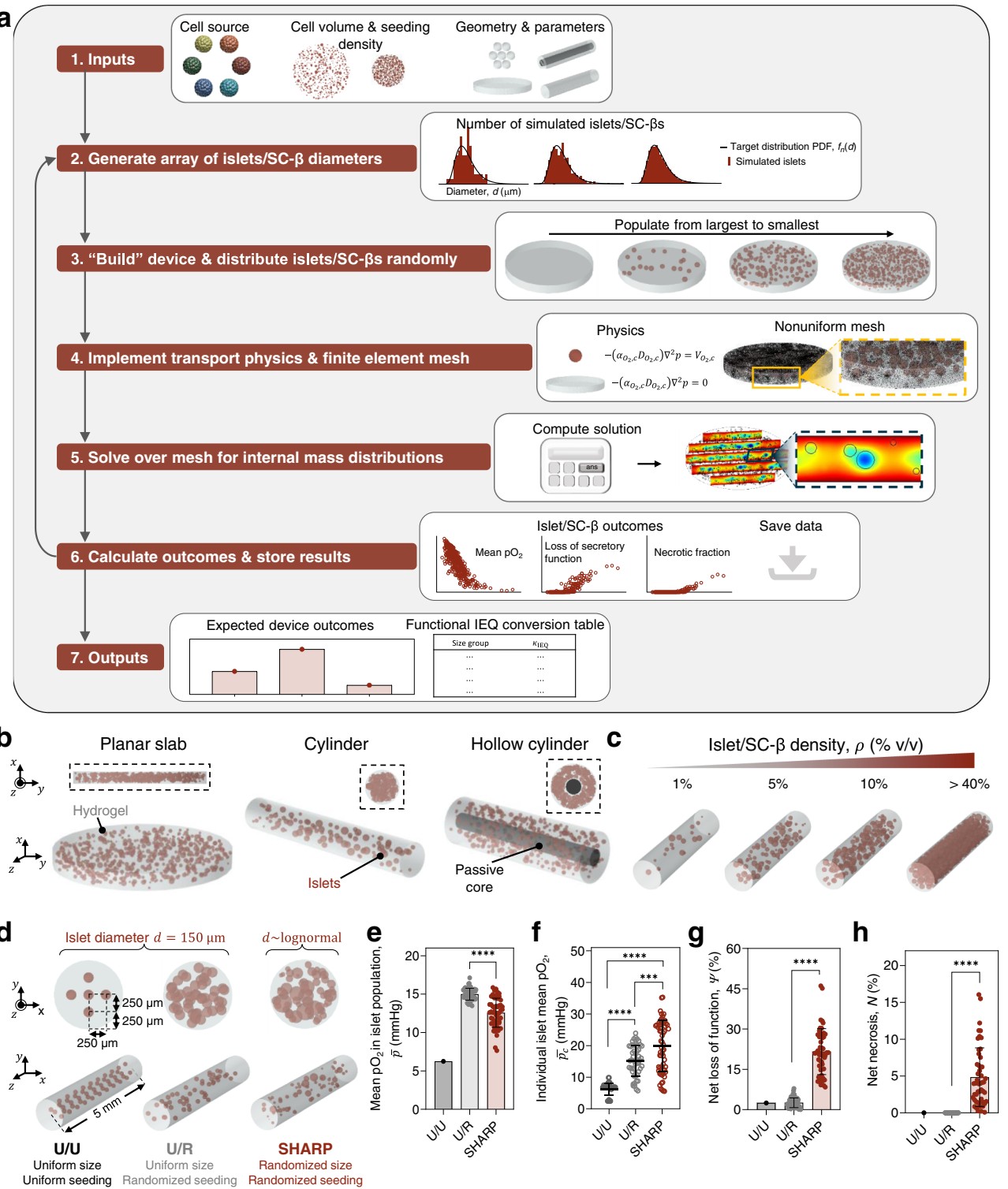

**Fig. 2 | SHARP enables the evaluation of a diverse range of bioartificial pancreas devices. a** Flowchart illustrating SHARP's operations. **b** Schematics displaying examples of device configurations, including the planar slab, cylinder, and hollow cylinder geometries. **c** Illustration of variable cell densities (as a percent of total volume) accommodated by SHARP in an example cylindrical system. **d** Visual representation of devices containing equivalent islet volumes with variable assumptions regarding the islet sizes and seeding positions: uniformly sized and uniformly spatially distributed islets (U/U), uniformly sized and randomly spatially distributed islets (U/R), and lognormal size distributed and randomly spatially distributed islets (SHARP). **e**–**h** Model results depend on size and seeding assumptions: mean islet population pO$_2$ ($\bar{p}$) (**e**), individual islet mean pO$_2$ ($\bar{p}_c$) (**f**), net loss of insulin secretion potential ($\Psi$) (**g**), and net necrotic percentage ($N$) (**h**) are significantly dependent on size and seeding assumptions. All data are presented as mean ± SD. **e** ****$p = 8.0e{-}11$, **g** ****$p = 6.9e{-}18$, and **h** ****$p = 6.9e{-}18$ (U/R versus SHARP, $n = 50$ iterations), unpaired two-sided Mann–Whitney test. **f** ***$p = 0.0008$ (U/R [$n = 50$ simulated islets] versus SHARP [$n = 62$ simulated islets]); ****$p = 2.0e{-}19$ (U/U [$n = 50$ simulated islets] versus SHARP); ****$p = 1.3e{-}17$ (U/U versus U/R); Brown–Forsythe and Welch ANOVA tests with two-sided Games–Howell post hoc $p$ value adjustment for multiple comparisons. U/R and SHARP data for **f** were collected from one iteration selected at random. Source data are provided as a Source Data file.

expected fractional loss of insulin secretion capacity relative to the maximum ability under fully oxygenated conditions, the latter two both functions of the local pO$_2$ (Supplementary Fig. 9c and Supplementary Eqs. (S14)–(S21)).

A comparison of islet size and spatial distribution assumptions conveys the importance of their careful implementation in mass transfer models of BAP devices. Three hypothetical cylindrical devices (1 mm diameter) are considered, each containing an equivalent islet volume (and cell density, $\rho = 2.3\%$ v/v) of rat islets with one of the following assumptions. All islets are 150 μm in diameter (the conventional definition of a standard islet representing one islet equivalent, IEQ), uniformly distributed in the hydrogel matrix (U/U). All islets are 150 μm in diameter, but randomly distributed (U/R). Islet diameters are randomly sampled from the empirical lognormal distribution and randomly distributed in the hydrogel matrix (SHARP) (Fig. 2d). Expected outcomes vary substantially according to the size and seeding assumptions. For example, uniform size assumptions U/U and U/R predict a substantially different mean islet pO$_2$ ($\bar{p}$) of 6.3 and 15.0 ± 0.8 mmHg, respectively, whereas SHARP predicts a mean pO$_2$ in between these values (12.6 ± 1.9 mmHg) (Fig. 2e). Further analysis reveals that the volume-average pO$_2$ within individual islets ($\bar{p}_c$) is predicted to range widely in SHARP even compared to assumption U/R, indicating that oxygenation variance between the islets is attributable to stochastic features of both sizes and spatial positions (Fig. 2f).

More critically, both uniform size assumptions (U/U and U/R) severely underestimate the more important downstream effects of hypoxia on cell survival and insulin secretion. For example, assumptions U/U and U/R suggest >8-fold lower levels of the expected loss of function compared to SHARP (2.5% versus 21.6 ± 8.5%) (Fig. 2g, h). In addition, both uniform size assumptions predict negligible necrosis ($N = 0\%$), whereas SHARP predicts that necrosis could vary up to ~15% of the total islet volume (despite assumption U/U predicting a significantly lower mean islet pO$_2$). An analysis of expected outcomes within individual clusters indicated that poor oxygenation, loss of function, and necrosis were all more severe in the largest islets (Supplementary Fig. 10). It was therefore unsurprising that adjusting the size of the islets of assumption U/R to diameters of 100 μm ($\bar{d}_n$) or 200 μm ($\bar{d}_v$) also underpredicted the extent of negative outcomes, especially necrosis (Supplementary Fig. 11). These findings reveal the sensitivity of models to assumptions regarding the size and spatial positions of the islets and affirm that recapitulating the full range of islet sizes and treating their positions as probabilistic are essential for obtaining unbiased predictions.

## Size distribution uncertainty significantly influences BAP device function

As model predictions are sensitive to the assumed islet diameters, with smaller islets being favorable (Supplementary Figs. 10 and 11)[10,27], it may be extrapolated that the properties of the encapsulated islet size distribution are relevant for device success (Fig. 3). Indeed, the extent of diabetes correction has been correlated to islet size properties in non-encapsulated intraportal islet transplant recipients in the clinic[28], and smaller[29–31] or reaggregated (and smaller)[32–34] islets have been reported to yield better outcomes in preclinical investigations. Though it is logical to presume that smaller islets may have better outcomes due to oxygen constraints, it is not evident if the variance in isolated islet size profiles is sufficient to significantly influence BAP graft outcomes. Thus, using SHARP, we explored the degree of sensitivity of expected device performance to islet size distributions within the range of their natural expected variability (Fig. 3a).

First, the size distributions of individual human islet isolations ($n = 129$, from the Alberta Diabetes Institute IsletCore at the University of Alberta) were characterized. A substantial variance in their size distributions was observed (Fig. 3b and Supplementary Fig. 12a), with the best-fit number-basis lognormal shape parameter ($\alpha_n$) ranging

from 0.20 to 0.56 and scale parameter (also the median diameter, $\beta_n$) from 85.1 to 154.0, respectively (Fig. 3c). Both $\alpha_n$ and $\beta_n$ were essentially normally distributed and uncorrelated (Supplementary Fig. 12b–d). Thus, a hierarchical probabilistic Monte Carlo model was developed to simulate a generalization of the distribution variance, producing traces that qualitatively resembled those of the empirical ones (Fig. 3d and Supplementary Fig. 12e–g).

We considered a representative cylindrical construct (1 mm diameter) containing 500 IEQ human islets at a density of 6% (v/v), simulated 200 times with distributions determined independently each iteration by sampling from the hierarchical probabilistic Monte Carlo model (Fig. 3e). The expected influence on device outcomes was significant: loss of insulin secretion capacity ($\Psi$) ranged from <1% to 32.5% and necrotic volume fraction from 0% up to 10.5% (Fig. 3f–i and Supplementary Fig. 13). Interestingly, the mean diameter ($\bar{d}_n$) was only weakly predictive of loss of function (Pearson's correlation coefficient, $r = 0.64$) (Fig. 3f). The islet size index (ISI), defined as the IEQ divided by the number of islets, is a commonly used characterization for generally describing the size profile of an islet isolation[28]. In this study, the ISI was reasonably correlated with graft outcomes ($r = 0.83$) (Fig. 3g). However, other parameters of the size distribution were more predictive, such as the standard deviation ($r = 0.95$) (Fig. 3h), and the diameter representing the third quartile of the volume distribution ($r = 0.93$) (Fig. 3i), among various other indices of the distribution's right tail weight (Supplementary Table 6). While the ISI has the advantage of being a simple calculation, these other metrics, derived from fitting a curve to the islet size frequencies, may be more reliable indicators of graft success. An evaluation of the individual islets from all iterations again revealed the size dependence of islet outcomes. All simulated islets >300 μm in diameter were at least partially necrotic ($N_c > 0$) and the insulin secretory capacity of those >250 μm in diameter was partially impaired ($\Psi_c > 0$) (Fig. 3j, k). We may define the functional islet equivalent volume (fIEQ) as the IEQ adjusted by the functional capacity of the cell cluster:

$$\text{fIEQ} = (1 - \Psi) \cdot \text{IEQ} \tag{3}$$

Plotting the fIEQ with respect to the IEQ for individual islets reveals a deviation of expected islet functional volume with an increasing diameter (Fig. 3l). Consequently, accounting for islets by volume alone may overstate their therapeutic contribution, especially of the larger ones. Loss of function of individual clusters was also dependent on the radial position, with more centrally located islets exhibiting higher impairment (Supplementary Fig. 14). Strikingly, we also found that small islets near large ones showed lowered oxygenation ($\bar{p}_c$) (Fig. 3m, n and Supplementary Fig. 15), indicating that large islets also restricted oxygen flow to smaller islets nearby. We repeated this study but constrained the maximum simulated islet diameter to 300 or 250 μm. Simply excluding larger islets significantly reduced expected loss of function from up to 32.5% to, at most, 16.8% and 10%, respectively (Fig. 3o), and necrosis levels from up to 10.5% to, at most, 1.6% and 0.4%, respectively (Fig. 3p).

The most salient finding from this analysis is that the performance of a primary islet-containing device is sensitive to uncontrollable uncertainty in the islet size distribution, and that this may be minimized by the exclusion of larger islets. The translational implication of this is that islet dosing in a BAP device may need to be adjusted according to the size distribution of the donor isolation. Likewise, in addition to inconsistency in diabetes induction methods[35], flawed IEQ accounting[36], and variable islet potency based on donor characteristics[37], isolation protocols[38,39], encapsulation processes[40], or downstream effects from combinations thereof[41], size distribution variability is another factor that may account for the well-documented irreproducibility[42] of device performance in preclinical animal models. Furthermore, these results advise that both the size and the polydispersity of SC-βs should be

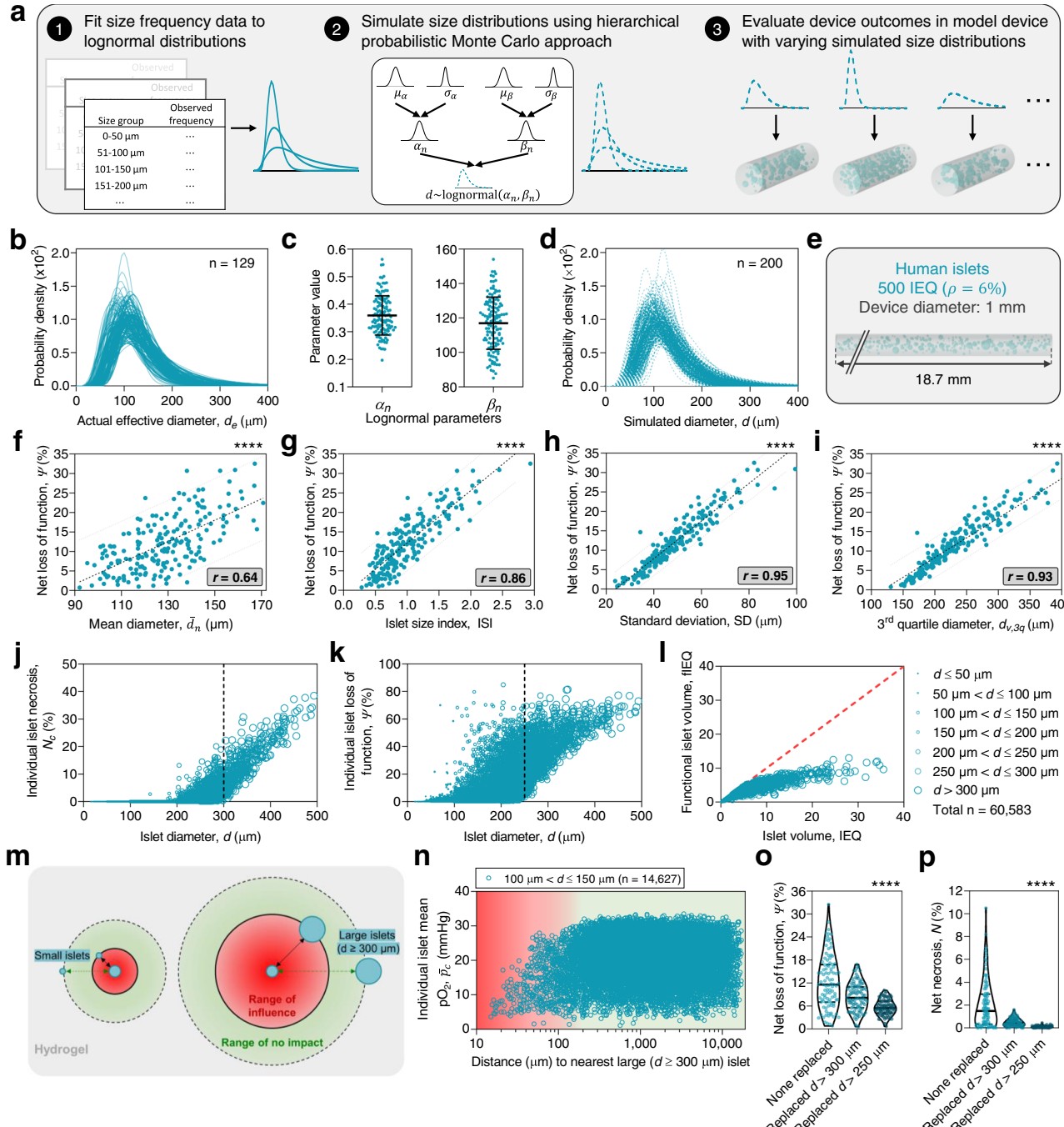

**Fig. 3 | Variability in human islet size distributions influences functional outcomes. a** Schematic illustrating the procedure for calculating the effect of size distribution variability on BAP device functional outcomes. **b** Overlaid traces of human islet diameter probability density functions ($n = 129$ distributions). **c** Values of the number-basis shape and scale parameters ($\alpha_n$ and $\beta_n$, respectively) from **b** ($n = 129$ values; bars represent mean ± SD). **d** Traces of simulated human islet diameter probability density functions generated through the hierarchical probabilistic Monte Carlo model ($n = 200$ simulated distributions). **e** Schematic of the simulated cylindrical system containing islets of size distributions selected by the hierarchical probabilistic Monte Carlo model. **f**–**i** Correlations of the loss of insulin secretion capacity ($\Psi$) to the following distribution properties: mean diameter ($\bar{d}_n$) (**f**), islet size index (ISI) (**g**), standard deviation (SD) (**h**), and volume-basis 3rd quartile diameter (**i**) ($n = 200$ simulated devices). Center dashed lines indicate the best-fit line and peripheral dashed lines indicate 95% confidence intervals. Pearson's correlation coefficient, $r$, is displayed in the bottom right text. **j**, **k** Necrotic fraction ($N_c$) (**j**) and fractional loss of insulin secretion capacity ($\Psi_c$) (**k**) of individual

simulated islets ($n = 60{,}583$ simulated islets). **l** Islet volume (IEQ) versus functional islet volume (fIEQ) for all individual islets. Line of identity is shown in red.
**m** Schematic showing that large islets reduce oxygenation of smaller ones localized nearby. **n** Mean islet $pO_2$ ($\bar{p}_c$) versus proximity to nearest large islet ($n = 14{,}627$ simulated islets). **o**, **p** Net loss of function (**o**) and necrosis (**p**) of the islet populations in the simulated construct when islets of all sizes were encapsulated ("none replaced", $n = 200$ simulated devices), or when the ones over 300 or 250 μm were excluded and replaced with smaller ones ($n = 100$ simulated devices). Solid black lines indicate median value, dashed black lines indicate first and third quartiles. **f** ****$p = 3.5e{-}25$, **g** ****$p = 1.1e{-}59$, **h** ****$p = 1.2e{-}100$, **i** ****$p = 4.0e{-}88$; two-sided Pearson's correlation. **o** ****$p = 5.7e{-}6$ ("none" versus ">300 μm replaced"), ****$p = 3.4e{-}15$ ("none" versus ">250 μm replaced"); **p** ****$p = 1.5e{-}10$ ("none" versus ">300 μm replaced"), ****$p = 1.1e{-}14$ ("none" versus ">250 μm replaced"); Brown–Forsythe and Welch ANOVA tests with two-sided Games–Howell post hoc $p$ value adjustment for multiple comparisons. Source data are provided as a Source Data file.

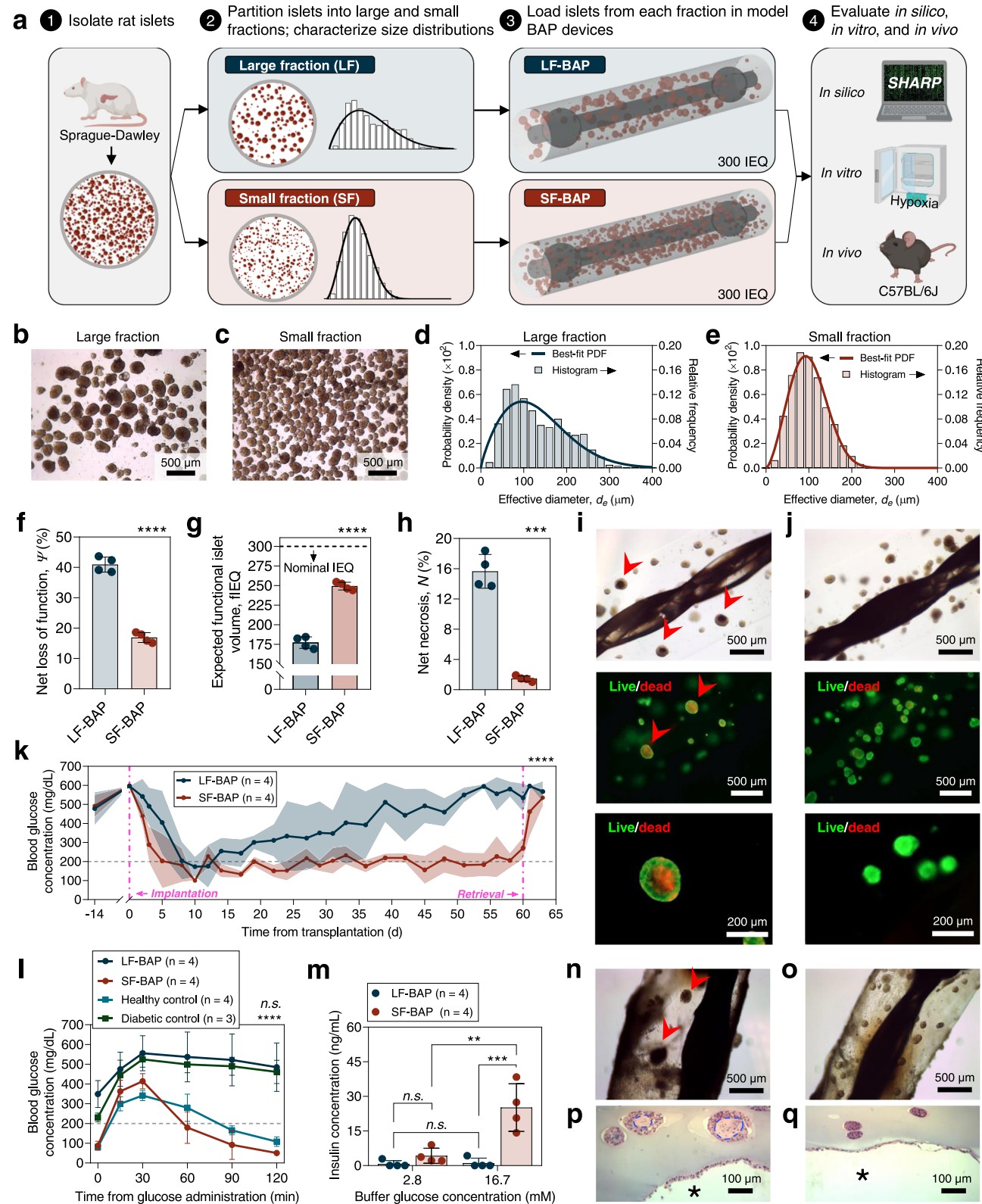

minimized as much as can be achieved. Finally, this analysis indicates several more sophisticated metrics, provided by islet size frequency curve fitting, which may be more predictive of graft success than the ISI.

## Size distributions affect the curative potential of BAPs in a diabetic animal model

We endeavored to validate the prediction that differences in the encapsulated size distribution significantly influence device

performance (Fig. 4). Sprague-Dawley rat islets were isolated and divided into two fractions. Then, most islets >250 μm in diameter were transferred from one fraction (the "small fraction") to the other (the "large fraction"). An equivalent volume of islets from each fraction was then encapsulated in identical BAP devices. The expected performance was evaluated using SHARP and then the physical devices were subjected to either hypoxic incubation or implantation in diabetic C57BL/6J mice (Fig. 4a).

**Fig. 4 | Islet size distributions affect the curative capacity of BAP devices in diabetic mice. a** Schematic describing the experiment. **b, c** Stereo microscope images from large (**b**) and small (**c**) fractions. **d, e** Histograms and best-fit diameter probability density functions of the large (**d**) and small (**e**) fractions. **f**–**h** SHARP predictions of fractional loss of function ($\Psi$) (**f**), functional islet volume (fIEQ) (**g**), and necrosis ($N$) (**h**), of large fraction- and small fraction-containing devices (LF-BAP and SF-BAP, respectively; $n = 4$ simulated devices). **i, j** Optical microscope (top) and live/dead (middle/bottom) images of one LF-BAP and SF-BAP following hypoxic incubation (arrows indicate necrosis). **k** Non-fasting BG concentrations of C57BL6/J mice before, during, and after LF-BAP ($n = 4$) or SF-BAP ($n = 4$) implantations. **l** IPGTT: BG readings versus time from glucose administration in healthy control mice (HC; $n = 4$), diabetic control mice (DC; $n = 3$), or mice receiving LF-BAPs ($n = 4$) or SF-BAPs ($n = 4$). **m** GSIS: insulin concentration in buffer after ex vivo incubation of LF-BAPs ($n = 4$) or SF-BAPs ($n = 4$) in 2.8 and 16.7 mM glucose. **n, o** Optical microscope images of representative retrieved LF-BAPs (**n**) and SF-BAPs (**o**). Arrows indicate necrosis. **p, q** Representative microscope images of H&E-stained slides of sections from the LF-BAPs (**p**) and SF-BAPs (**q**). Dashed outlines indicate regions with severe karyorrhexis and loss of nuclei. Asterisks indicate the host side of the device-host boundary. Data are presented as mean ± SD. **f, g** ****$p = 3.5e-6$ (unpaired two-sided Student's $t$-test). **h** ***$p = 0.0008$ (unpaired two-sided Student's $t$-test with Welch's correction). **k** ****$p = 1.3e-22$; one-way ANCOVA. **l** n.s. ($p = 0.7453$, LF-BAP versus DC; $p = 0.9991$, SF-BAP versus HC); ****$p = 3.8e-10$ (HC versus LF-BAP), ****$p = 3.5e-7$ (HC versus DC), ****$p = 2.0e-8$ (SF-BAP versus LF-BAP), ****$p = 2.5e-6$ (DC versus SF-BAP). **m** **$p = 0.0011$ (SF-BAP, 2.8 versus 16.7 mM); ***$p = 0.0030$ (16.7 mM, LF-BAP versus SF-BAP); n.s. ($p > 0.9999$, LF-BAP, 2.8 versus 16.7 mM; $p = 0.9481$, 2.8 mM, LF-BAP versus SF-BAP); two-way ANOVA with two-sided Sidak's post hoc $p$ value adjustment. Source data are provided as a Source Data file.

Characterizations of the rat islet isolation fractions confirmed that the partitioning produced a significant difference in the size distributions (Fig. 4b, c and Supplementary Fig. 16a, b). For comparison, islets with $d_e$ >200 μm represented 64.7% of the large fraction islet volume but only 27.0% of that of the small fraction. It was also determined that the $d_e$ distributions of both fractions could be well described by the Weibull function (Fig. 4d, e and Supplementary Fig. 16c, d), enabling a priori evaluation by SHARP. As desired, distribution metrics predictive of graft outcomes, such as the SD and ISI, all favored the success of the small fraction (Supplementary Fig. 16e).

A BAP device previously reported by our group[43] was selected as a model encapsulation system for comparing performance when containing an equivalent mass of large fraction islets (LF-BAPs) or small fraction islets (SF-BAPs). The device is comprised of a twisted suture dip-coated with a layer of nano-porous poly(methyl methacrylate) (PMMA), the latter of which facilitates the robust attachment of a concentric layer of islet-encapsulating alginate hydrogel to the twisted thread (Supplementary Fig. 17a). The curative potential of any cell therapy is dependent on the dose of cells delivered. Conventionally, an islet volume of 500 IEQ is used in this rat-to-mouse model, however lower cell doses may achieve diabetes correction if a higher degree of cell survival can be attained[35]. We therefore elected to encapsulate an islet volume of 300 IEQ (in ~16 μL alginate) to increase the sensitivity of mouse diabetes correction to the functional capacity of the encapsulated cells.

The rat islet-containing BAP devices were modeled in SHARP, essentially as a hollow cylinder with the twisted suture representing the passive core (0.5 mm diameter) and the islet-encapsulating hydrogel (0.5 mm thick) representing the concentric layer (Supplementary Fig. 17b). Significantly better performance of devices containing islets sourced from the small fraction was predicted (Supplementary Fig. 17c, d). For example, the loss of insulin secretion capacity was expected to be 40.9 ± 2.5% in the LF-BAPs versus 16.9 ± 1.6% in the SF-BAPs (Fig. 4f). This implies that the LF-BAPs only contained the functional equivalent (fIEQ) of ~177 IEQ, in contrast to ~249 IEQ in the SF-BAPs (Fig. 4g). In addition, necrosis was expected to exceed 15% of the islet volume within LF-BAPs in contrast to a minimal ~1% in the SF-BAPs (Fig. 4h). SHARP predicted that the magnitude of necrosis within individual islets would be related to the islet size, with the largest islets (>350 μm in diameter) exceeding 50% necrotic and those <200 μm exhibiting negligible or zero necrosis (Supplementary Fig. 17e, f). Qualitatively, these predictions were corroborated for devices incubated for 2 days in hypoxic (5% oxygen, ~40 mmHg) conditions similar to levels expected in the in vivo milieu. Live/dead staining revealed that several larger islets in the LF-BAP exhibited a dead core surrounded by a live periphery, whereas virtually all islets in the SF-BAP device showed no signs of cell death (though a small amount of core cell death was observed in a few of the relatively larger islets) (Fig. 4i, j).

Alternatively, LF-BAPs and SF-BAPs were transplanted intraperitoneally in STZ-induced diabetic C57BL/6J mice and evaluated for 60 days. The blood glucose (BG) levels of mice treated with SF-BAPs lowered to ~200 mg dL⁻¹ within 3–4 days and remained low for the duration of the 60-day implant, whereas those of the mice treated with LF-BAPs gradually elevated to >450 mg dL⁻¹ over this period (Fig. 4k). Likewise, the percent change in body weights of mice treated with SF-BAPs gradually increased over the 60-day period whereas those receiving LF-BAPs only increased for the first 30 days before flatlining following this initial period, corresponding to the loss of glycemic control (Supplementary Fig. 18). An intraperitoneal glucose tolerance test (IPGTT), wherein mice were fasted and then administered a glucose bolus, was conducted on day 58. Mice receiving SF-BAPs showed restored normoglycemic BG levels (<200 mg dL⁻¹) within 120 min, slightly delayed in comparison to those of the healthy control mice, whereas those receiving LF-BAPs showed elevated BG levels over 120 min, similar to those of the diabetic control mice (Fig. 4l).

After retrieval, the BGs of all mice elevated to over 450 mg dL⁻¹, confirming the role of the devices in diabetes correction. In addition, a static glucose-stimulated insulin secretion (GSIS) test with the explanted devices showed significantly higher glucose responsiveness of the SF-BAPs than LF-BAPs (for the latter of which, 3 of 4 exhibited negligible insulin secretion) (Fig. 4m). Confirming these findings, ex vivo stereo microscope imaging showed that most of the islets in LF-BAPs were dark with rough surfaces, indicative of cell death, whereas most in SF-BAPs were light yellow with smooth surfaces, suggestive of maintained viability (Fig. 4n, o). Hematoxylin and eosin (H&E) staining corroborated this observation, showing many completely viable islets in SF-BAPs, and mostly completely denucleated islets in LF-BAPs, with some large ones showing the characteristic pattern of a nonviable core surrounded by a viable periphery as predicted by SHARP (Fig. 4p, q and Supplementary Figs. 19 and 20). H&E staining confirmed that the one LF-BAP with detectable glucose responsiveness upon GSIS contained several partially viable islets (Supplementary Fig. 19d). It furthermore appeared that fibrotic coverage was generally more substantial upon LF-BAPs, though quantification of the number of fibrotic cell layers on the surface of retrieved devices did not indicate statistical significance ($p = 0.0844$) (Supplementary Fig. 21).

The prediction that device function is significantly dependent on the islet size distribution, with smaller, less polydisperse ones being favored was generally supported by the results of this study. With respect to expected functional capacity, the magnitude of the difference was perhaps underpredicted by SHARP, given the nearly complete loss of function in some LF-BAPs. One possibility is that the greater fibrotic coverage in most LF-BAPs suffocated the grafts. In addition, because islet OCR is correlated to local glucose conditions, SHARP indicated that the survival and function of the encapsulated rat islets would be reduced if exposed to high glucose, which may be expected in the hyperglycemic LF-BAP recipients. We speculate that, in addition to well-documented deleterious effects of metabolic

exhaustion resulting from chronic hyperglycemia as functional cell mass is gradually lost over time[44], elevated BG levels due to poor graft function may induce sustained increases in islet OCR, in turn worsening islet function and survival and may thus accelerate graft attrition in the mouse model (Supplementary Fig. 22). Alternatively, or in addition, this may suggest that the broader negative effects of necrosis in stimulating a more aggressive immune response by DAMP release or by increasing graft suffocation by immune cells attracted to the device surface are substantial. Finally, this experiment represents a general application of the SHARP-enabled method for predicting the performance of islet delivery devices, whereby the islet size distribution is first characterized, and then the expected function can be evaluated by this platform prior to transplantation.

## Optimizing clinical-scale BAP devices and determining curative islet doses

SHARP was next used to optimize curative capacity devices in a human patient with regards to cell loading density and geometry (Fig. 5). It is estimated that 500 k fIEQ (~0.88 cm³) of intraportally engrafted human islets are required to restore normal glucose regulation in an average individual (Supplementary Eq. (S24))[45]. Though this volume is relatively small, the islets must be encapsulated at low volumetric densities and in thin structures to mitigate the oxygen limitation-related effects of reduced insulin secretion capacity and survival rate[46]. Increasing cell loading density or device thickness allows for more cells to be contained in a smaller device footprint, but at diminishing or even negative returns due to oxygen-limited efficacy.

In this study, we considered a curative dose as the islet volume with the equivalent insulin secretion potential of 500 k IEQ fully functional islets (i.e., 500 k fIEQ), and varied the characteristic device thickness and cell density for three generic device shapes: the planar slab, the cylinder, and the hollow cylinder. The objective of the optimization was to minimize the characteristic curative device size under the constraint of some maximum amount of necrosis, which we call the necrotic tolerance ($N_t$). Because it is not well characterized how much cellular necrosis is permissible without inducing serious externalities, the optimal device, $S'_{N_t}$, was calculated under the constraint of three necrotic tolerances, volume fractions of 5%, 10%, and 20% of the total islet volume. In addition, we report the actual volume of islets, $IEQ_{cure}$, required to achieve the curative functional dose for the optimal devices.

Estimates for a planar slab exposed to the host tissue on both faces containing human islets surprisingly suggested a quite feasible size of the optimal device. For planar constructs, the characteristic size is best represented by the diameter of a circular slab containing a curative islet dose ($D_{cure}$). Slab thickness ($\tau$) was varied between 0.5 and 1.5 mm and volumetric human islet density ($\rho$) between 2% and 20%. Heat maps were generated to illustrate the relationship between the inputs ($\tau$ and $\rho$, the horizontal and vertical axes, respectively) and the response variables ($D_{cure}$ and $N$, shown as a colorimetric value for each combination of the inputs). Interestingly, the curative diameter showed a complex relationship with cell density and slab thickness (Fig. 5a and Supplementary Fig. 23). Increasing the cell density reduced the curative diameter at any slab thickness. On the other hand, necrosis clearly increased with both increasing the cell density or slab thickness, however at a less drastic rate for thinner constructs. It is revealing to plot the curative diameter at the maximum allowable cell density conditional on the expected necrosis being below each necrotic tolerance, at each tested slab thickness (Fig. 5b). Logically, relaxing the necrotic tolerance from 5% to 10% or 20% allowed the system to support higher cell densities at each slab thickness thus yielding smaller constructs. However, the magnitude of this advantage decreased with decreasing slab thickness. This paradoxically resulted in the curative diameter being minimized by reducing the slab thickness because the benefits to islet function and survival at lower slab

thicknesses outweighed the geometrical benefit of increasing the thickness. For the planar configuration, there was one optimal device, S', which satisfied all three necrotic tolerance ($N = 1.2 \pm 0.5\%$) and is represented by the highest volumetric cell density ($\rho = 20\%$) and lowest slab thickness ($\tau = 0.5$ mm) of the variable space tested. It is estimated that this optimal device could contain a curative islet dose of 628 k IEQ (Fig. 5c) within a circular slab of 11.9 cm in diameter and 0.5 mm in thickness (i.e., with a footprint of 111.2 cm²). While the fabrication of such a device may be difficult, the directional implications of this analysis are clear: in planar constructs, cell survival and function are very sensitive to increases in thickness; thus, it should be minimized, while the cell density should be increased.

Less optimistic results were obtained for human islets in a cylindrical construct. Here, the characteristic size is best represented by the length, $L_{cure}$. Cylinder diameter ($D$) varied between 0.5 and 1.5 mm and cell density between 2% and 20%. Both outcome parameters were monotonically related to both input variables though in opposing directions: the curative length ($L_{cure}$) was minimized by increasing both the cell density and cylinder diameter, but this increased the expected necrotic fraction (Fig. 5d and Supplementary Fig. 24). Unlike planar constructs, the benefits of reducing the diffusion distance by reducing the cylinder diameter did not outweigh the geometrical advantages. Plotting the curative length at the maximum cell density for each necrotic tolerance constraint at each tested cylinder diameter showed that increasing the cylinder diameter reduced the curative length to a certain extent, above which negative returns were realized (Fig. 5e). Under the most conservative necrotic tolerance ($N_t = 5\%$), the optimal device, $S'_5$ (described by $\rho = 14\%$ and $D = 1$ mm), had an expected curative length of 12.8 m. The optimal devices under relaxed necrotic tolerances of 10% and 20% ($S'_{10}$ and $S'_{20}$) still suggest that 9.1 and 7.5 m of the cylinder would be required to deliver a curative dose, respectively. Moreover, while relaxing the necrotic tolerance yielded a reduction in the curative length, it required a higher curative dose: the projected curative cell dose ($IEQ_{cure}$) of $S'_5$, $S'_{10}$, and $S'_{20}$ were 713 k, 886 k, and 1035 k IEQ, respectively (Fig. 5f). It is unlikely therefore that any cylindrical construct may be conceivably transplanted in a human recipient.

A BAP device with an overall cylindrical shape may nonetheless be desirable for ease of transplantation, fabrication, or other factors. The hollow cylinder was thus considered, containing human islets with a fixed hydrogel thickness of 0.5 mm and a variable acellular core diameter ($D_i$) between 1 and 5 mm (and cell density between 2% and 20%) (Supplementary Fig. 25). Both outcome parameters of this system were essentially monotonic across both input variables in the same directions as those of the cylinder, though necrosis showed a weaker dependence on the inner diameter (Fig. 5g). Results clearly suggested increasing the inner diameter as much as possible for reducing the curative length, though the returns of this strategy were diminishing and at the expense of increasing the total device volume (Fig. 5h). The optimal devices, $S'_5$, $S'_{10}$, and $S'_{20}$ were all at the maximum tested inner diameter of 5 mm with maximum allowable cell densities of 8%, 13%, and 18% of the total graft volume, respectively. However, this increase in permittable cell density only reduced the curative length from 2.0 to 1.6 and 1.4 m, respectively, with associated curative islet volumes of 771 k, 1051 k, and 1274 k IEQ (Fig. 5i). Furthermore, the expected loss of function ($\Psi$) was relatively constant for inner diameters larger than 3 mm at any given cell density; thus, we may imagine the implications of expanding the inner diameter. At an inner diameter of 12 mm, incorporating a human islet density of 18% would yield a curative length of ~64 cm, which is significantly more manageable than the meter-scale estimates above. It was previously demonstrated that removing the largest islets improves graft function. Estimates for a hollow cylinder of fixed inner diameter of 5 mm suggested that a notable reduction in the curative length may be achieved by removing islets >200 μm, in proportion to the conservatism of the necrotic

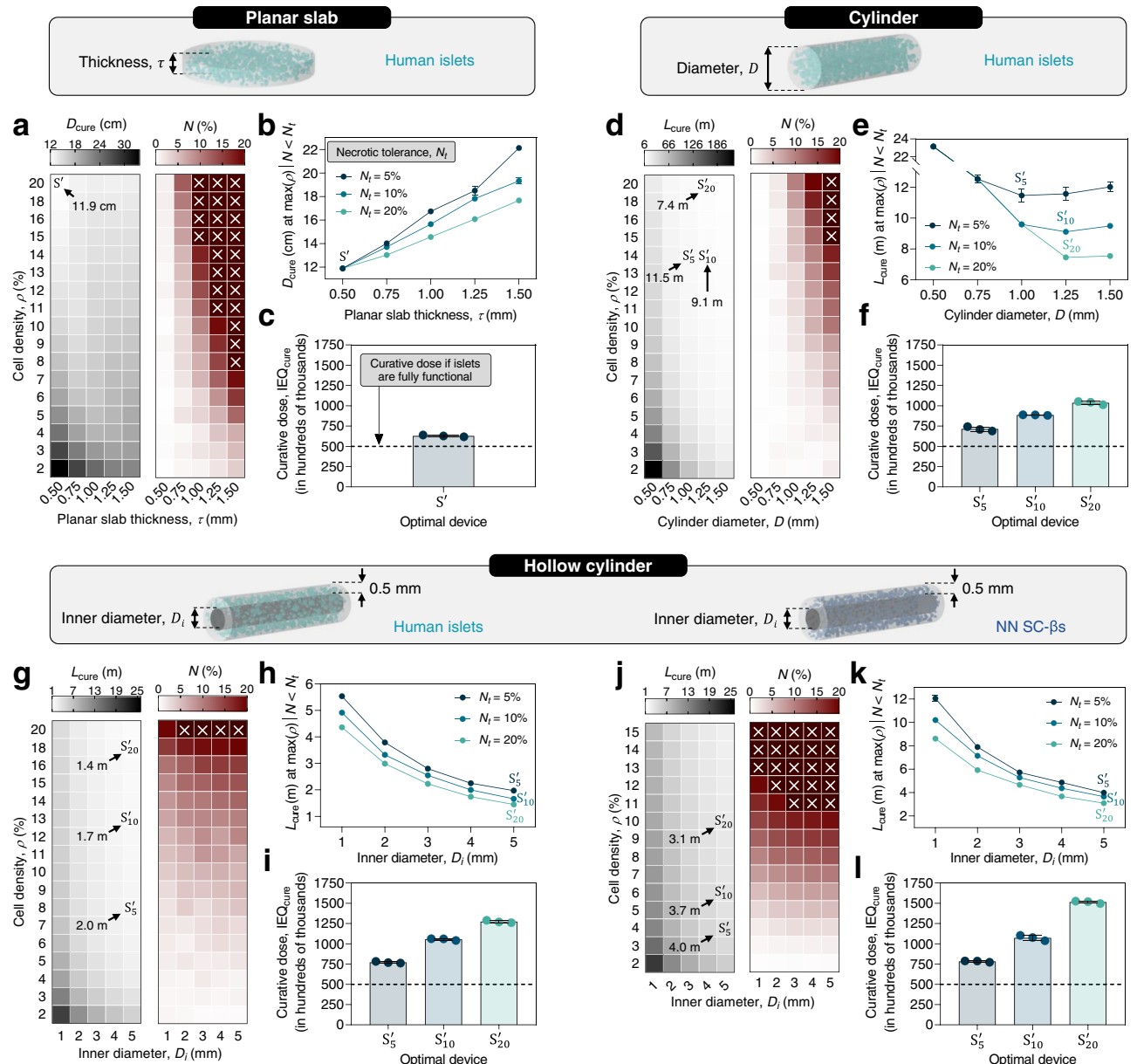

**Fig. 5 | Optimizing devices for delivering a curative islet payload to a human patient.** **a**–**c** Optimization of planar slab devices with human islets. Heat maps (**a**) showing the curative device diameter ($D_{cure}$, left) and necrotic percentage ($N$, right), displayed as a colorimetric value represented by the top-positioned scale bar, for all combinations of the input variables slab thicknesses ($\tau$; horizontal axis) and volumetric cell density ($\rho$; vertical axis). Curative slab diameter ($D_{cure}$) at the maximum cell density permissible within the necrotic tolerance ($N_t$) constraints 5%, 10%, and 20%, at each tested slab thickness (**b**). Estimated volume of islets (IEQ$_{cure}$) required to deliver 500 k fIEQ in optimal device S' (**c**). **d**–**f** Optimization of cylindrical devices with human islets. Systems $S'_5$, $S'_{10}$, and $S'_{20}$ represent optimal cylindrical devices for necrotic tolerances of 5%, 10%, and 20%, respectively. Heat maps (**d**) showing the curative device length ($L_{cure}$, left) and necrotic percentage ($N$, right) for all combinations of

combinations of the input variables cylinder diameter ($D$) and cell density ($\rho$). Curative device length ($L_{cure}$) at the maximum cell density for necrotic tolerances 5%, 10%, and 20% at each cylinder diameter (**e**). Estimated curative dose (IEQ$_{cure}$) for optimal devices $S'_5$, $S'_{10}$, and $S'_{20}$ (**f**). **g**–**l** Optimization of hollow cylindrical devices with human islets (**g**–**i**) and NN SC-βs (**j**–**l**). Heat maps (**g**, **j**) showing the curative device length ($L_{cure}$, left) and necrotic percentage ($N$, right) for all combinations of the input variables hollow cylinder inner diameter ($D_i$) and cell density ($\rho$). Curative device length ($L_{cure}$) at the maximum cell density for necrotic tolerances ($N_t$) of 5%, 10%, and 20%, at each inner diameter (**h**, **k**). Estimated curative dose (IEQ$_{cure}$) for optimal devices $S'_5$, $S'_{10}$, and $S'_{20}$ (**i**, **l**). White crosses in **a**, **d**, **g**, and **j** indicate values overrange. Data in **b**, **c**, **e**, **f**, **h**, **i**, **k**, and **l** are presented as mean ± SD ($n$ = 3 iterations). Source data are provided as a Source Data file.

tolerance (e.g., a length reduction of >50% was expected under a strict necrotic tolerance of $N_t$ = 1%, versus length reductions of 14.2% for $N_t$ = 5% and 6.2% for $N_t$ = 20%; Supplementary Fig. 26).

It may therefore be expected that the optimal systems of grafts containing SC-βs would be more realizable given their more symmetrical and generally smaller size distributions. However, even if we generously assume that the absolute insulin secretion rate is equivalent to that of human islets, more pessimistic estimates for both curative size

and dose were obtained because of their relatively higher OCR. An analysis of NN SC-β-containing hollow cylinder devices showed a similar directional relationship between the outcome parameters and the input variables (Fig. 5j and Supplementary Fig. 27). Optimal systems $S'_5$, $S'_{10}$, and $S'_{20}$ were found at the maximum tested inner diameter, 5 mm, but tolerated much lower cell densities of 4%, 6%, and 10%, respectively, corresponding to curative lengths of 4.0, 3.7, and 3.1 m (Fig. 5k). Likewise, the expected curative dose of an average patient was projected to

be higher than that of human islets for each necrotic tolerance at 782 k, 1073 k, and 1515 k IEQ, respectively (Fig. 5l).

There are several revealing implications of the analysis discussed above. First, estimates of the curative dose for all constructs except the optimal slab were quite high. The average yield of human islet isolation is roughly 300 k ± 180 k IEQ[47]. The suboptimal function of the encapsulated islets in optimized systems therefore requires that multiple donors (and devices) may be needed for full glycemic restoration. Therefore, the requirement of a greater cell mass for SC-βs may not be of significant consequence as one of their primary advantages is their ability to be produced in virtually unlimited quantities. Furthermore, the poor viability and performance of islets in these constructs indicates the advantage of technologies that generate or provide exogenous oxygen[48–50] or improve oxygen transport by other means[51,52].

### A machine learning model as a surrogate for SHARP

We have demonstrated that SHARP can calculate expected outcomes for a diverse set of cell sources and programmable devices. However, it is based on resolving numerical solutions to partial differential equations using the stochastic finite element method, which is computationally burdensome and utilizes proprietary software which may require expert use or novice training[53]. Overcoming these constraints would greatly increase its versatility and ease of use (Fig. 6 and Supplementary Fig. 28). With this aim, we developed an ensemble machine learning model (SHARP-ML), which, once trained, could reproduce SHARP's predictions accurately and within seconds without specialized software or expertise (Fig. 6a).

We first established a proof-of-concept application for SHARP-ML. Conventionally, islet volume in an isolation is measured by counting the number of islets in 50 µm diameter-ranged bins, and, through volumetric equivalence coefficients ($\kappa_{IEQ}$), converting to IEQ following the customary definition of 1 IEQ equaling the volume of a sphere with a diameter of 150 µm[54]. A convenient standardized volume metric has proven incredibly useful, but, as we have shown, it may overstate the therapeutic contribution of encapsulated islets, especially larger ones, to an extent that is dependent on the properties of the delivery device (Supplementary Fig. 29). An adjustment to $\kappa_{IEQ}$ values based on expected functional capacity, which we denote as $\kappa_{fIEQ}$, may more accurately reflect the therapeutic potential of cell clusters in each size group.

Accordingly, SHARP-ML was trained to data collected from human islet systems evaluated in the optimization study (Supplementary Fig. 30). Because it is rare to select a well-performing model a priori[55], six machine learning models were selected, providing a range of functional approaches to learn the associations between $\kappa_{fIEQ}$ and the predictor variables. The models included three tree-based methods (LightGBM[56], XGboost[57], and Cubist[58] rules), one instance-based approach (kernel K-nearest neighbors[59]), a neural network model, and a simple linear model. The result was given by the average results of each constituent model (Supplementary Fig. 31) weighted by Pearson's correlation coefficients of the fit to the training data (Supplementary Fig. 32), allowing the ensemble to leverage the unique strengths of each model for different subgroups of the parameter space.

It was validated that $\kappa_{fIEQ}$ predictions by SHARP-ML resembled those calculated by SHARP (Supplementary Fig. 33). This may be demonstrated specifically by comparing $\kappa_{fIEQ}$ predictions from SHARP and SHARP-ML for two hypothetical systems: (1) a planar slab with a thickness $\tau = 0.55$ mm and human islet density $\rho = 2.5\%$, and (2) a cylinder with a diameter $D = 1.3$ mm and $\rho = 14.5\%$. SHARP-ML's $\kappa_{fIEQ}$ predictions for both constructs were in close agreement with those calculated by SHARP (Fig. 6b and Supplementary Fig. 34). For the hypothetical slab, $\kappa_{IEQ}$ and $\kappa_{fIEQ}$ coefficients were quite similar for all size groups except the largest two, indicating that loss of function in this system would be expectedly exclusively concentrated in the largest islets (Supplementary Fig. 34a). Much lower functional

contributions were expected for islets in the hypothetical high-density cylindrical construct: $\kappa_{fIEQ}$ for the size group $d > 350$ µm was predicted to have an fIEQ of ~3, whereas islets in this size group have an IEQ of ~15.8 (Supplementary Fig. 34b). This >5-fold difference highlights the potential difference between the measured islet volume and its equivalent functional capacity, especially in highly stressed systems.

The value of the machine learning model is understood when comparing the ease of obtaining $\kappa_{fIEQ}$ coefficients: SHARP required several hours (on a workstation with a base clock speed of 2.11 GHz and a total random access memory of 15.8 GB), whereas SHARP-ML produced tabulated coefficients within seconds on a web app or on an offline database (Fig. 6c and Supplementary Fig. 35; it is hosted at https://worland.shinyapps.io/sharp-ml/). Though it is, at present, limited to evaluating devices containing human islets and configured in one of the three conventional geometries explored herein, SHARP-ML can be easily updated to consider a wider variety of systems by training it to more data via the method presented above. In summary, this tool enables researchers to make quick comparisons between the expected efficacy of different designs without requiring expertise in mass transfer or access to significant computational resources.

## Discussion

Herein we described a computational platform for predicting the oxygenation and related outcomes of islets/SC-βs in generic BAP devices, which accounts for the variance of the cell cluster sizes and uncertainty regarding their spatial distributions. After determining the size distributions of many common cell sources, we demonstrated that accounting for these properties in mass transfer models is necessary for obtaining unbiased results. We then found that variance in the size distributions of human islets should result in significant variability in their function when encapsulated, and that this may be reduced for the benefit of device performance by excluding the largest islets. This finding was validated in diabetic mice, where islets of artificially manipulated size distributions encapsulated at the same volume density in model devices showed variable efficacy depending on their size distribution. An optimization study then found the preferable use of planar or hollow cylinder devices for delivering a therapeutic cell payload and demonstrated the ability to calculate the expected curative cell dose. Finally, an app-hosted machine learning model robustly approximated results from the stochastic finite element model overcoming the computational burdens of the latter approach and enabling its use for a wider audience of non-experts.

SHARP can be used to evaluate general hypothetical systems as demonstrated in the optimization studies but is more effective when provided inputs for specific BAP devices (e.g., as in the analysis of the model constructs tested in mice). In addition, considerations of stochasticity herein were limited to the geometrical properties of the devices, though some other aspects of the physical system are probabilistic in nature. First, the oxygen tension at the device surface was fixed at a constant 40 mmHg, an intermediate value in the range of reported measurements in common transplantation sites and a value conventionally used in other deterministic approaches (Supplementary Table 1). The value of this parameter has a notable effect on model predictions, and if known in a particular system, may be adjusted in SHARP to obtain estimates at specific oxygen tensions (Supplementary Fig. 36). Likewise, we considered all devices to use low concentration alginate hydrogel as the encapsulation matrix. However, a variety of hydrogel materials and compositions have been explored for this purpose, which each may be endowed with oxygen permeabilities differing from the value used here. A sensitivity analysis of the diffusion coefficient of oxygen in hydrogel in a test construct suggests that a change in this parameter yields a roughly one-to-one change in the mean pO$_2$ of the islets over a reasonable range, thus, when modeling a specific device, accurate implementation of this parameter is indeed important (Supplementary Fig. 37). Furthermore, SHARP considers all islet/SC-β

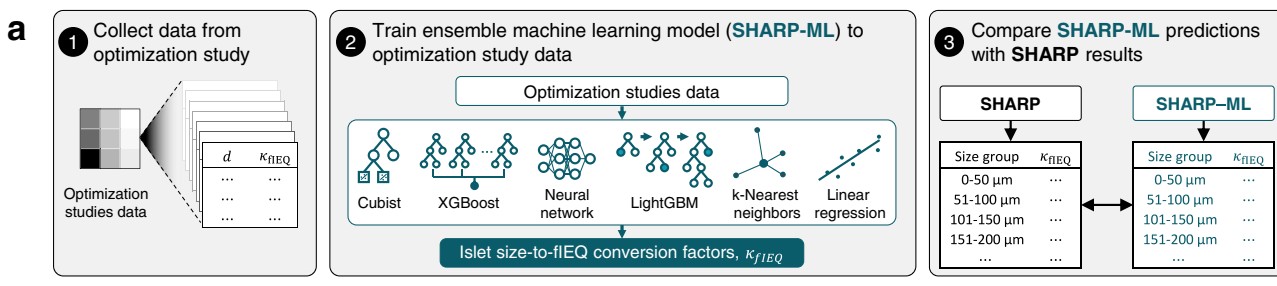

**b** Example size-to-fIEQ conversion factor ($\kappa_{\text{fIEQ}}$) table.

| Islet size group ($d$, μm) | $\kappa_{\text{IEQ}}$ (Conventional coefficients) | Hypothetical device 1 | | Hypothetical device 2 | |
|---|---|---|---|---|---|
| | | $\kappa_{\text{fIEQ}}$ (SHARP) | $\kappa_{\text{fIEQ}}$ (SHARP-ML) | $\kappa_{\text{fIEQ}}$ (SHARP) | $\kappa_{\text{fIEQ}}$ (SHARP-ML) |
| < 50 | - | 0.028 | 0.028 | 0.022 | 0.022 |
| 51–100 | 0.167 | 0.170 | 0.170 | 0.124 | 0.126 |
| 101–150 | 0.648 | 0.564 | 0.574 | 0.380 | 0.385 |
| 151–200 | 1.685 | 1.485 | 1.520 | 0.834 | 0.854 |
| 201–250 | 3.500 | 3.146 | 3.174 | 1.203 | 1.374 |
| 251–300 | 6.315 | 5.488 | 5.400 | 1.514 | 1.724 |
| 301–350 | 10.352 | 7.537 | 7.807 | 1.950 | 1.949 |
| > 350 | 15.833 | 11.090 | 11.831 | 2.596 | 2.960 |

Hypothetical device 1 — Planar slab — Human islets $\rho$ = 2.5% — $\tau$ = 0.55 mm
Hypothetical device 2 — Cylinder — Human islets $\rho$ = 14.5% — $D$ = 1.3 mm

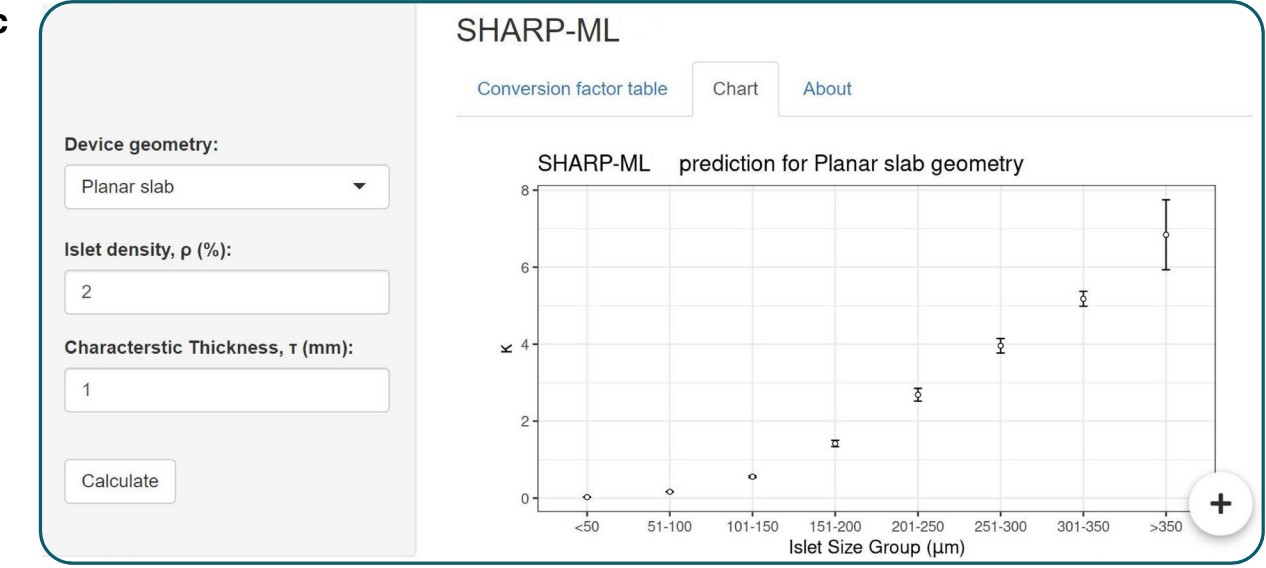

**Fig. 6 | Ensemble machine learning model achieves rapid prediction of function adjusted IEQ conversion coefficients. a** Procedure for developing and validating ensemble machine learning model (SHARP-ML) to predict size group-to-fIEQ conversion coefficients ($\kappa_{\text{fIEQ}}$): (1) individual islet results from the optimization studies (Fig. 5) were collected; (2) an ensemble of machine learning models were trained to subsets of the optimization studies data, producing $\kappa_{\text{fIEQ}}$ coefficients according to results from each constituent model weighted by Pearson's correlation, $r$, to the training data; (3) SHARP-ML predictions were cross-validated with SHARP calculations for hypothetical encapsulation systems not included in the training dataset. **b** Example islet size group-to-fIEQ conversion table for two hypothetical encapsulation systems, showing $\kappa_{\text{fIEQ}}$ calculations by SHARP and predictions SHARP-ML. The conventional volumetric size group-to-IEQ coefficients, $\kappa_{\text{IEQ}}$, are shown for comparison in the second column. **c** Screenshot from SHARP-ML web application ($n = 18$ summarized model predictions): users can operate the left panel to indicate the device geometry and cell density, producing the $\kappa_{\text{fIEQ}}$ coefficients on the right panel. Data in the inset plot are presented as mean ± SD ($n = 18$ model predictions summarized from >$2 \times 10^7$ observations). Source data are provided as a Source Data file.

tissue between and within sources as equal with respect to their physical and biological composition (except for variable OCRs between sources). This may be a better assumption for SC-βs than for primary islets, the latter of which have been shown to have subpopulations with distinct respiration and insulin secretion rates[60]. Finally, many properties such as the sensitivity of insulin secretion capacity to local oxygen are variable even within sources according to factors as subtle as incubation time[61]. Pending their more robust characterization, the uncertainty of these parameters may be integrated into this computational platform to obtain further generalized results.

We can nonetheless make general conclusions from the analyses performed herein. It was demonstrated throughout that insulin secretion capacity is highly sensitive to the parameters of the BAP device and even the size distribution of the encapsulated islets. Therefore, the islet volume (i.e., IEQ) may be inadequate as a dosage standard of therapeutic potential. To account for this, we proposed the adjusted metric fIEQ and conversion coefficients $\kappa_{fIEQ}$, which define standards for the functional capacity of the encapsulated cells. Because the goal of any device is to deliver insulin to the host circulation system rather than solely maintain the mass of the encapsulated cells, such a metric may more accurately reflect the therapeutic potential of a BAP device. SHARP provides a computational framework to make such calculations.

Moreover, SHARP is viable to be integrated with other systems to greatly improve its applicability. For example, in this work, curve fitting of size distributions was performed separately. A sensible update would be to configure the program to accept isolation images, fractional counts in the conventional size groups, or information from a cell cluster counter device[62] and automate the distribution curve fitting prior to mass transfer analysis. This would enable end-to-end analysis without reliance on custom user inputs or the use of generalized size distributions. Furthermore, recently developed islet-scale models of insulin dynamics[9-11] can be easily incorporated into SHARP to consider insulin action at the device scale and under stochastic treatment. Likewise, recently, an FDA-approved simulator has demonstrated the ability to model glycemic levels in simulated patients following selected routes of insulin administration (NCT03093636)[63]. The integration of SHARP with this program would represent a closed-loop quantitative representation of insulin dynamics in a clinically transplanted BAP device.

We also demonstrated that an app-hosted ensemble machine learning model could be a successful surrogate for the stochastic finite element model. The proof-of-concept application pursued here enabled outcome predictions to be obtained in a fraction of the time and computational burden, with high accuracy. Additional training of the machine learning model to a wider range of input data (e.g., other cell sources, device designs, etc.) and for other outcomes (e.g., cell death) could vastly expand its scope. The success of this approach for the application tested herein suggests that it may be a viable method for estimating results from other nonlinear stochastic systems traditionally modeled by finite element analysis as well.

In summary, statistical and numerical methods were leveraged together to model the expected performance of therapeutic-secreting cell delivery devices, accounting for critical uncertainties inherent to such systems. Notable model predictions were validated by in vitro and in vivo experimentation. Ultimately, this work represents a flexible platform for predicting and optimizing the function of cell delivery devices, which may aid the clinical translation of cell encapsulation therapies for T1D.

## Methods
All research complied with relevant ethical regulations and were approved by the Cornell Institutional Animal Care and Use Committee.

### Chemicals
Sodium chloride (NaCl), calcium chloride dihydrate (CaCl$_2$·2H$_2$O), barium chloride dihydrate (BaCl$_2$·2H$_2$O), PMMA, dimethylformamide (DMF), and D-glucose were purchased from Sigma-Aldrich. Ultrapure sodium alginate (Pronova SLG100) was purchased from NovaMatrix. Water was deionized to 18.2 mΩ·cm with a Synergy UV purification system (Millipore Sigma).

### Animals
Two-month-old male C57BL/6J mice were purchased from The Jackson Laboratory and were maintained at a temperature of 70–72 °F with 30–70% humidity under a 14-h light/10-h dark cycle. Male Sprague-Dawley rats (weighing ~300 g) were purchased from Charles River Laboratories and were maintained at a temperature of 70–72 °F with 30–70% humidity under a 12-h light/12-h dark cycle. All animal procedures were approved by the Cornell Institutional Animal Care and Use Committee and complied with relevant ethical regulations (Protocol # 2012-0144).

### Islets and SC-βs
Juvenile porcine islets were purchased from the Professor Jonathan Lakey group at the University of California, Irvine. NN SC-βs, from their research protocol, were generously provided by Novo Nordisk (Måløv, Denmark). WU SC-βs were generously provided by the Professor Jeffrey Millman group at Washington University in St. Louis. Human islet information was collected from the Alberta Diabetes Institute IsletCore database at the University of Alberta[64]. Rat and mouse islets were isolated by our lab with the procedures described below.

Rat islets were harvested from Sprague-Dawley rats (~300 g). The rats were anesthetized using 3% isoflurane in oxygen throughout the surgical procedure. In brief, the pancreas was distended with 10 mL 0.15% Liberase (Roche) in M199 media (Gibco) through the bile duct. The pancreas was digested at 37 °C in a circulating water bath for ~28 min (digestion time varied slightly for different batches of Liberase). The digestion was stopped by adding cold M199 media with 10% heat-inactivated fetal bovine serum (FBS; Gibco). After vigorously shaking, the digested pancreases were washed twice with media (M199 + 10% FBS), filtered through a 450-μm sieve, and then suspended in a Histopaque 1077 (Sigma)/M199 media gradient and centrifuged at 1700 × g with 0 break and 0 acceleration for 17 min at 4 °C. This gradient centrifugation step was repeated for higher purity. Finally, the islets were collected from the gradient and further isolated by a series of gravity sedimentations, in which each top supernatant was discarded after 4 min of settling. Islets were then washed once with islet culture media (RPMI 1640 supplemented with 10% FBS, 10 mM Hepes, and 1% penicillin/streptomycin) and cultured in this medium overnight before imaging.

Mouse pancreatic islets were isolated from 8-week-old male C57BL/6J mice. One bottle of Collagenase (CIzyme RI 005-1030, VitaCyte) was reconstituted in 30 mL M199 media and put on ice. Mice were sacrificed and the intraperitoneal space was exposed using scissors. Cannulation through the bile duct was performed using a small animal butterfly catheter infusion set (#72-5967, Harvard Apparatus) followed by perfusion with cold Collagenase as described previously[65]. The perfused pancreases were then removed and digested in a 37 °C water bath for 21 min. The digestion was stopped by adding 20–25 mL of cold M199 media with 10% FBS and slight shaking. After digestion, the same purification protocol described above for rat islets was followed.

### Characterizations
Optical and fluorescent images were taken using a digital inverted microscope (EVOS FL) with the EVOS AMF4300 imaging system. High magnification H&E staining images were taken by an Aperio Scanscope (CS2) using ISCapture 3.9. Stereo microscope images of mouse islets, BAP devices, and low magnification images of H&E-stained slides were obtained using stereo microscope (Olympus SZ61) with ISCapture 3.9. Images of WU SC-β batches were generously provided by the Millman group. ImageJ 1.52p software was used for image analysis. For

histological characterizations, devices were fixed in 10% formalin, dehydrated with graded ethanol solutions, embedded in paraffin, and sectioned by the Cornell Histology Core Facility. Samples were sliced on a microtome at a thickness of 5 μm and stained with H&E. For viability determinations, cells were stained with the LIVE/DEAD Viability/Cytotoxicity Kit (Invitrogen).

## Mass transport theory

Salient model features are schematized in Supplementary Fig. 9. The BAP systems modeled consisted of two or subdomains, the hydrogel matrix (denoted by subscript $h$) and the encapsulated cell clusters (denoted by subscript $c$), and exclusively in the case of the LF-BAPs and SF-BAPs, the nylon thread (denoted by subscript $t$). Oxygen is poorly soluble in aqueous media such as hydrogels and tissue, thus it may be appropriately described as a diluted species with a conservation equation given by

$$\frac{\partial c}{\partial t} = D_{O_2,i}\nabla^2 c - V_{O_2,i} \tag{4}$$

where $c$ is the dissolved oxygen concentration (in mol m$^{-3}$), $t$ is time (in seconds), $D_{O_2,i}$ is the diffusion coefficient of oxygen in subdomain $i$ (in m$^2$ s$^{-1}$), $\nabla^2$, the Laplacian operator, is the second derivative with respect to all spatial dimensions (e.g., $\frac{\partial^2}{\partial x^2} + \frac{\partial^2}{\partial y^2} + \frac{\partial^2}{\partial z^2}$ in Cartesian coordinates), and $V_{O_2,i}$, a function of $c$, is the volumetric OCR in subdomain $i$ (in mol m$^{-3}$ s$^{-1}$). Henry's law allows us to relate the dissolved oxygen concentration to the partial pressure ($p$, in mmHg), which is conveniently uniform across the interfaces between the cell clusters and the hydrogel

$$c = \alpha_{O_2,i} p \tag{5}$$

where $\alpha_{O_2,i}$ is the partial pressure-dependent solubility coefficient of oxygen in subdomain $i$ at 37 °C (in mol m$^{-3}$ mmHg$^{-1}$).

We are concerned with calculating the oxygen distribution within BAPs several weeks or months after transplantation. The time scale ($\hat{t}$) for relaxing gradients within a diffusion-driven system is approximated by

$$\hat{t} \sim \frac{L^2}{D_{O_2,e}} \tag{6}$$

where $L$ is the characteristic diffusion length (in m) and $D_{O_2,e}$ is the effective diffusion coefficient of the system (in m$^2$ s$^{-1}$). Considering the most conservative case of a slab with a half-thickness of 1 mm and an order of magnitude estimate for the effective diffusivity of $1.0 \times 10^{-9}$ m$^2$ s$^{-1}$, we find that the equilibration time is on the order of 1000 s (or ~17 min). As $\hat{t}$ is significantly lower than our time of interest, we considered the system at a steady state. Accordingly, the transient term of Eq. (4) may be eliminated, and the relationship described in Eq. (5) may be substituted into Eq. (4) to derive the species conservation equation in terms of $p$

$$\left(\alpha_{O_2,i} D_{O_2 i}\right)\nabla^2 p = V_{O_2,i} \tag{7}$$

Following literature consensus[9–12,66,67] based on studies in isolated rat liver mitochondria[68], respiration in the cell clusters ($V_{O_2,c}$) was governed by Michaelis–Menten kinetics and a step-down function attributed to the lack of oxygen consumption in necrotic cells (Supplementary Fig. 9b):

$$V_{O_2,c} = \begin{cases} 0, & p < p_N \\ \frac{p V_{max}}{p + K_V}, & p \geq p_N \end{cases} \tag{8}$$

where $V_{max}$ is the maximum basal OCR (in mol m$^{-3}$ s$^{-1}$), $K_V$ is the half-maximal coefficient (in mmHg), and $p_N$ is the threshold pO$_2$ (in mmHg) required for cell survival. No oxygen consumption occurred in the hydrogel ($V_{O_2,h} = 0$), nor, in the case of the LF- and SF-BAPs, the thread ($V_{O_2,t} = 0$).

Oxygen-dependent insulin secretion potential (Supplementary Fig. 9c), S, was defined only in the cell clusters and modeled according to the Hill relationship[9–11], in favor of the bilinear[12,67] or polynomial[13] formulations used by others, based on studies of isolated islets exposed to variable media oxygen levels[8]

$$S = \begin{cases} 0, & p < p_N \\ \frac{p^{n_S}}{p^{n_S} + (K_S)^{n_S}}, & p \geq p_N \end{cases} \tag{9}$$

with half-maximal coefficient $K_S$ (in mmHg) and Hill coefficient $n_S$ (unitless). The step-down function again represents the loss of insulin secretion capacity of necrotic cells. More precisely, $S$ defines the relative second-phase insulin secretion rate, rather than insulin concentration, and is hence not defined in the hydrogel. Accordingly, the volume-average insulin secretion potential of the cluster, $\bar{S}_c$, was given by

$$\bar{S}_c = \frac{1}{V_c} \iiint_C S\, dV \tag{10}$$

where $C$, the integral region, is the cell cluster and $V_c$ is the volume of the cell cluster. We defined and reported the loss of insulin secretion potential of the cell cluster, $\Psi_c$, simply as

$$\Psi_c = 1 - \bar{S}_c \tag{11}$$

and the loss of insulin secretion of the entire population of encapsulated islets is calculated as the weighted average of the individual islets, where the weights are each islet's volume.

Three geometries were considered for the shape of the BAP (Supplementary Fig. 9): the planar slab, the cylinder, and the hollow cylinder. For simulations of the cylindrical and hollow cylindrical geometries, no-flux ($\alpha_{O_2,h} D_{O_2,h} \nabla p = 0$) conditions were imposed on the ends, and a constant pO$_2$ was implemented on the lateral face; for simulations of the planar slab geometry, a no-flux condition was implemented on the lateral face, whereas the constant pO$_2$ ($p = p_{ext}$) was implemented on the ends (Supplementary Fig. 9a). For analyses of the LF-BAP and SF-BAP devices, the internal thread was implanted as a uniform nylon material; for analyses of the hollow cylinder in the optimization study (Fig. 5g–l), we considered a system where the passive core material consisted of the same hydrogel as the encapsulation matrix. In both cases, flux continuity was imparted at the interface between the passive core and encapsulation hydrogel. Finally, we assumed that both the hydrogel and cell clusters were uniform with respect to all physical parameters. Calculations for the mean pO$_2$ ($\bar{p}$), loss of insulin secretion capacity ($\Psi$), and net necrotic percentage ($N$), as well as a description of the model implementation, are provided in the Supplementary Materials and methods. Model parameters are produced in Supplementary Table 7.

## Development of SHARP-ML

Six machine learning models were selected as constituents of an ensemble to provide a range of functional approaches (i.e., model structures) to learn the associations between $\kappa_{flEQ}$ values and the predictor variables (Fig. 6a). These included LightGBM (3.3.2.)[56], XGboost (1.6.0.1)[57], Cubist[58] rules, kernel K-nearest neighbors (v1.1.14)[59], a simple linear model, and a neural network model. The neural network model

selected was a multi-layer perceptron with a rectified linear activation function (ReLU) with the following variables: (1) number of hidden layers and (2) hidden units in each layer that were selected during the hyperparameter tuning process. Prior to model training, the training data were a subset for each geometry, and the hyperparameters for each model were tuned for each subset. The training data consisted of the $\kappa_{\text{fIEQ}}$ values generated using SHARP in the optimization study (Fig. 5). For each of the underlying machine learning models, the Pearson correlation coefficient was estimated for different subgroups of the data (defined for each geometry in Supplementary Fig. 32) and was used as weights for each constituent model's contribution to the final ensemble. More information is provided in Supplementary Materials and methods in subsections Methods machine learning model development and Supplementary Data 1–5.

### Rat islet-containing BAP device fabrication

BAP devices were fabricated according to our previously reported procedure[43]. Briefly, a suture (Ethilon nylon suture, 5–0, monofilament, Ethicon, Inc.) was twisted, folded, and knotted at the ends. This structure was then submerged into a PMMA/DMF solution (7% w/v) which contained $CaCl_2$ (2.5% w/v) for ~3 s, after which it was air dried for 24 h and sterilized by UV exposure or a hydrogen peroxide plasma sterilizer. Using the conventional conversion table, the total rat islet yield obtained after partitioning was calculated at ~1658 and ~1698 IEQ for the large and small fractions, respectively, to which ~88 and ~91 µL of 2% w/v alginate was added such that the volumetric islet density was equivalent in both groups. The modified threads were then inserted into a polyethylene tube (1.5 mm inner diameter), which was then filled with the rat islet-containing alginate suspension (300 IEQ in ~16 µL 2% w/v alginate) and allowed to crosslink for 5 min. The devices were then carefully pushed out from the tube using a pipet tip and submerged into a crosslinking buffer containing 95 mM $CaCl_2$ and 5 mM $BaCl_2$. Finally, the devices were rinsed three times in saline (0.9% w/v NaCl in water) before transferring to islet culture medium.

### Implantation and retrieval of BAP devices

Devices were implanted into the peritoneal cavity of 8-week-old male immunocompetent C57BL/6J mice. To induce diabetes, healthy mice were administered an intraperitoneal injection of freshly prepared 22.5 mg mL$^{-1}$ (in 100 mM sodium citrate buffer, pH ~4.5) STZ (Sigma-Aldrich) solution at a dosage of 150 mg STZ per kg mouse body weight. The BG levels of all mice were over 585 mg dL$^{-1}$ 14 days after STZ administration. To implant the devices in the intraperitoneal space, the diabetic mice were first anesthetized using 3% isoflurane in oxygen and their dorsal skin was shaved and sterilized using betadine and 70% ethanol. Then, a lateral transverse incision (~2 mm) was made on the midline of the abdomen and the peritoneum was exposed using blunt dissection. The peritoneum was then grasped with forceps and an incision (~2 mm) was made, exposing the peritoneal cavity. The devices (one device per mouse) were then inserted through the incisions into the peritoneal cavity, and both the peritoneal incision and skin were closed using 5-0 nylon sutures (CP Medical). To retrieve the devices, the peritoneal cavity was exposed as described above and the devices were removed into a saline solution using forceps. The incisions were then sutured shut using 5-0 nylon sutures.

### Ex vivo static GSIS assay

Krebs-Ringer Bicarbonate (KRB) buffer (2.8 mM CaCl$_2$·2H$_2$O, 1.2 mM MgSO$_4$·7H$_2$O, 1.2 mM KH$_2$PO$_4$ 4.9 mM KCl, 98.5 mM NaCl, and 25.9 mM NaHCO$_3$, all from Sigma-Aldrich) was supplemented with 20 mM Hepes (Gibco) and 0.1% bovine serum albumin (Sigma-Aldrich). Retrieved devices were incubated in KRB buffer supplemented with 2.8 mM glucose for 2 h at 37 °C and 5% CO$_2$. Devices were then transferred and incubated in KRB buffer supplemented with 2.8 mM

glucose, and then 16.7 mM glucose for 75 min each. The buffer was collected after each incubation step. Insulin concentration was measured using an ultrasensitive rat insulin enzyme-linked immunosorbent assay kit (ALPCO).

### BG monitoring and IPGTT

BG levels in mice were measured using a commercial glucometer (Contour Next EZ, Bayer) by collecting a drop of blood from the tail vein. For the IPGTT, mice were fasted for 16 h and then administered an intraperitoneal injection of 20% glucose solution at a dosage of 2 g glucose per kg mouse body weight. BG levels were measured at 0, 15, 30, 60, and 120 min following glucose injection.

### Hypoxic incubation

A New Brunswick Galaxy CO170 incubator, which has dynamic control over carbon dioxide, nitrogen, and temperature levels, was used for hypoxic incubation of the rat islet-containing BAP devices. The incubator was equipped with compressed carbon dioxide and nitrogen gas cylinders, which controlled the gas mixture to 5% oxygen, 5% carbon dioxide, and 90% nitrogen (at a temperature of 37 °C) by modulating nitrogen inflow. BAP devices were placed in a 6-well plate containing islet culture medium at a buffer height of 2 mm (before BAP device addition), pre-equilibrated to the hypoxic gas mixture.

### Statistics

Data are expressed as raw values, mean ± SD, or mean ± interquartile range, as indicated in the figure legends. Pre hoc tests were performed on all results for determining normality (Shapiro–Wilk) and equality of variances (*F*-test) to establish which statistical method was appropriate. Comparisons between two groups across one factor were performed using the Mann–Whitney *U* test for data determined not to be normally distributed, the unpaired two-sided Student's *t*-test for data normally distributed, and the unpaired two-sided Student's *t*-test with Welch's correction for data normally distributed but with unequal variances. Comparisons between more than one group across one factor were performed using the one-way Brown–Forsythe and Welch analysis of variance (ANOVA) with the two-sided Games–Howell post hoc *p* value adjustment for multiple comparisons for groups with unequal variances or the traditional one-way ANOVA with Sidak's post hoc *p* value adjustment for multiple comparisons, otherwise. Random BG and percent change in body weight measurements during the rat islet-containing BAP studies were compared via an analysis of covariance where time was considered a continuous covariate and treatment (e.g., LF-BAP versus SF-BAP) was treated as a discrete factor. BG measurements during the IPGTT were compared via a two-way ANOVA with Sidak's post hoc *p* value adjustment for multiple comparisons where both time and treatment were considered discrete factors. Likewise, insulin concentrations in the buffer during the GSIS were compared via a two-way ANOVA with Sidak's post hoc *p* value adjustment for multiple comparisons. Correlations were determined by the two-sided Pearson's correlation test. In Fig. 4b, c, 1 representative image of 5 total collected images is shown for each group; in Fig. 4n, 1 of 10 is shown; in Fig. 4o, 1 of 12 is shown; in Fig. 4p, 1 of 77 is shown; in Fig. 4q, 1 of 117 is shown. Analyses were performed in GraphPad Prism 8 and R v4.1.1. Statistical significance was concluded at $p < 0.05$.

Additional methods can be found in the Supplementary Materials and methods. This includes the procedure for size distribution fitting, the hierarchical probabilistic Monte Carlo model for simulating human islet size distributions, SHARP model implementation, and SHARP-ML model development.

### Reporting summary

Further information on research design is available in the Nature Research Reporting Summary linked to this article.

## Data availability

All data supporting the findings of this study are available within the article and the supplementary information files and from the corresponding author upon reasonable request. Data collected from the Alberta Diabetes Institute IsletCore at the University of Alberta can be found by following registration instructions at https://www.epicore.ualberta.ca/isletcore/Default. Source Data are provided with this paper.

## Code availability

SHARP source code is freely available for download on GitHub (https://github.com/alexanderuernst/SHARP) or the corresponding Zenodo repository[69]. The R scripts used to prepare the data, train the surrogate machine learning model (SHARP-ML), and make the inferences can be freely accessed on GitHub (https://github.com/scworland/sharp-ml). The web application can be accessed at https://worland.shinyapps.io/sharp-ml/ (username: guest, password: sharp-ml). Source code for the web application can be downloaded freely on GitHub (https://github.com/scworland/sharp-ml-app).

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

## Acknowledgements

This work was partially supported by the National Institutes of Health (NIH, 1R01DK105967), the Novo Nordisk Company, the Juvenile Diabetes Research Foundation (JDRF, 2-SRA-2018-472-S-B), and the Hartwell Foundation (M.M.). This material is also based upon work supported by the National Science Foundation Graduate Research Fellowship under grant number DGE-1650441 (A.U.E.). Some schematics in Figs. 1a and 4a were created with BioRender.com. We thank the Cornell University Animal Health Diagnostic Center for histological sectioning and staining, the Alberta Diabetes Institute IsletCore at the University of Alberta for permitting the use of their human islet isolation data, and the Professor Millman group and Novo Nordisk for their generous provision of their respective stem cells and related information.

## Author contributions

A.U.E., L.-H.W., and M.M. designed and conceived the project. A.U.E. and A.K.D. formulated the mass transfer model and implementation. L.-H.W. and A.U.E. performed the rodent study. S.C.W., A.U.E., and S.S. formulated the statistical and machine learning models. L.-H.W., X.W., W.L., and A.C. performed rodent islet isolations. T.K. and D.O. performed clinical islet isolations. B.A.M.-G and A.M.J.S. analyzed and interpreted data from the clinical islet isolations. A.M.J.S. contributed to discussions on clinical relevancy. A.U.E. wrote the manuscript; L.-H.W., S.C.W., B.A.M.-G., A.K.D., K.K.P., and M.M. provided substantial edits.

## Competing interests

The authors declare no competing interests.
