## [Peer Review File · Nature Communications]

REVIEWER COMMENTS

Reviewer #1 (Remarks to the Author):

In this manuscript, authors developed a predictive computational platform for optimizing the design of encapsulation devices of pancreatic islets as well as the stem cell-derived β -cell sources. Statistical and numerical methods and machine learning were combined to develop the platform to consider the uncertainties in a typical bioartificial pancreas including the varying size distributions of islets and their random localization within the device. This platform is also scalable and expandable to the large scale device for clinical setting. This platform clearly and scientifically answers the critical questions of islet biologist, stem cell bioengineers and engineers for developing the implantable encapsulation devices of beta cells for diabetes, which could be a great tool for exploring future islet transplantation strategies. The manuscript is quite well written with robust data to support the authors' concept, though the content may not be targeting wide range of readers.

- 1) How is the hydrogel concentration considered for the in silico simulations, which critically affects the molecular diffusion? Authors used clinical-grade hydrogel but several products are available with different concentration and softness, which could be one of the essential variables to be included.
- 2) How did the authors integrate the O₂-dependency of insulin-secreting ability of islets embedded in the BAP device? This may be the important factor for the success of BAP device in regard to the oxygenation strategy.
- 3) In Fig.4, authors stated "An equivalent volume of islets from each fraction were then encapsulated in identical BAP devices." Since the use of "equivalent" volume (300 IEQ) in the preparation between 2 groups of large fraction and small fraction is critically important that directly affects the transplantation outcome shown in Fig.4k, authors may demonstrate the data of absolute IEQ according to the size categories used for the in vivo transplantations.
- 4) Fig.4 demonstrated the animal studies using cylinder-shaped BAP system, to validate the SHARP in silico simulations. Since authors insist that the SHARP was developed that can be applied to variable types of BAP systems including planner slab and hollow cylinder (which authors later introduced the optimal configurations in Fig.5 for clinical settings), validity of SHARP may be tested not only in the single BAP system but in multiple systems (e.g. Cylinder shaped and Planner slab).
- 5) I anticipate that the readers would have hard time to understand the heatmaps in Fig.5, although there are supplemental supporting materials (Suppl Fig.22). It would be great if authors could explain a bit more for better understanding in broad readers.

Reviewer #2 (Remarks to the Author):

The manuscript describes a computational model that aims to predict bioartificial pancreas device performance by using randomly placed and sized islets to make the model more representative of real-life implementations. The model predicts a significant influence of endogenous variance in islet size distributions on device performance, and the manuscript also includes some limited in vitro and in vivo validation for this (e.g., a cell encapsulation device with pre-selected smaller islets performed better in diabetic mice than the same device with larger islets). Based on this, it also proposes a size conversion tool to calculate effective islet equivalents based on the specifics of the device used. The manuscript is well-written and organized, and it includes considerable amount of supporting information to document details of the work performed. The work as is, is mainly of interest for those in the bioengineering and artificial organ field focused on beta cell replacement. Major and minor issues that need to be addressed are summarized in detail below.

Major Comments

1. The fact that smaller islets are likely to perform better than larger one when transplanted (directly or in a device) has been suggested by several other models before – some cited here (e.g., *Theor. Biol. Med. Model.* 2011, 8, 20), some not (e.g., *Artif. Organs* 2018, 42, E168). It is also why reaggregated islets of more uniform (and reduced) size are and have been explored for transplantation by several groups (e.g., *Diabetologia* 2010, 53, 937; *Tissue Eng Part A* 2013, 19, 604; *Diabetologia* 2018, 61, 2016). The novelty here is the implementation of the random distribution and its use to predict device-dependent islet equivalent conversion factors. Many of these considerations also hold for straightforward islet transplantation (i.e., not in a device) as those islet also rely on passive diffusion for several days and clustering is a likely reason for primary nonfunction in many cases. Maybe if this work could be extended to also incorporate such cases, e.g., at well vascularized sites such as the liver or kidney vs not well vascularized site such as subcutaneous or fat pad, this might make it of more general interest and not just for the relatively small community of islet transplantation in bioartificial pancreas devices. Regarding this, it would probably be nice to also mention some of the publications suggesting that small islets perform better in transplantation than larger ones, e.g., *Am. J. Physiol. Endocrinol. Metab.* 2006, 290, E771; *Diabetes* 2007, 56, 594; *Pancreas* 2020, 49, 650.

2. Regarding fitting of the size distributions (e.g., Supplementary Fig. 3, Supplementary Table S4), for (theoretical) consistency it might be better to use either Weibull or log-normal across all fits despite differences in shapes. Ultimately, fit is quite similar, and I understand the authors' desire to use the best fit, but it is hard to justify why would size distribution be a Weibull function for mouse but a lognormal for rat (and human) islets. I'd suggest picking one and using that uniformly across all, which would also allow more clear-cut comparisons, and just mention that the other one could be used as an alternate option with very similar result. On a related note, Supplementary Table 3 claims log normal as better fit based on RMSE. A quick recalculation with that data gave a much smaller difference for me (see below): RMSE of 0.0248 vs 0.0500 and not 0.0060 vs 0.0500 as shown there – please double check.

< 50 0.0662 0.0104
51-100 0.3319 0.3291 0.3339
101-150 0.4095 0.3993 0.4096
151-200 0.1537 0.1639 0.1645
201-250 0.0587 0.0207 0.0462
251-300 0.0253 0.0007 0.0117
301-350 0.0132 0.0000 0.0029
351-400 0.0078 0.0000 0.0008
RMSE 0.0500 0.0248 (vs. 0.0060 shown)

3. Effect of fibrosis on worsening hypoxia is not mentioned at all except that fibrotic cell coverage appeared to be more substantial for the device with larger islets (Supplementary Fig. 20). This could also significantly alter device performance after a few days, and the estimated IEQ calculations could be overestimates for devices likely to be exposed to more severe fibrotic coverage. One possible reason why the LF-BAP performed worse in vivo than predicted by the model. Also, exhaustion of remaining cells due to increasing metabolic demand as functional mass is being reduced over time is not included in the model – another possible reason for overpredicting performance.

4. For the modeling (e.g., mass transport) it is claimed that “All BAP systems modeled consisted of two subdomains, the hydrogel matrix (denoted by subscript h) and the encapsulated cell clusters (denoted by subscript c). What about devices with inert core, such as the nylon suture – what oxygen transport is assumed there? In Supplementary Fig. 16c,d, it looks like there is oxygen transport in the core nylon suture - is that true? I don't think that is realistic, maybe for the covering PMMA layer only.

5. The prediction of function implemented for adjusted IEQ conversion coefficients (e.g., Fig. 6) seems to be independent of species; if it is, why? Oxygen consumption rates are significantly higher in rodents (e.g., mice) than in humans (Table S5), so loss of function should be higher too - why is this not coming up in f_{IEQ} ?

Minor Comments

1. N/A

Reviewer #3 (Remarks to the Author):

The ML application is really interesting, since the computational time in this kind of application (several hours) is not acceptable. Moreover, the ML approach seems to improve the performances in the case of bigger size. Only few comments below to be consider before publication:

- the authors stated that they used a "neural network" (NN) model: several different types of NNs are available, which one has been used here? Also, the parameters used for the other techniques should be reported in the supplementary material.

In general, details to understand the ML methodologies are not reported and should be added.

- The results are related to SHARP-ML approach without defining in detail which approach between the 6 proposed is the optimal one and why.

Reviewer #1 (Remarks to the Author):

In this manuscript, authors developed a predictive computational platform for optimizing the design of encapsulation devices of pancreatic islets as well as the stem cell-derived β -cell sources. Statistical and numerical methods and machine learning were combined to develop the platform to consider the uncertainties in a typical bioartificial pancreas including the varying size distributions of islets and their random localization within the device. This platform is also scalable and expandable to the large-scale device for clinical setting. This platform clearly and scientifically answers the critical questions of islet biologist, stem cell bioengineers and engineers for developing the implantable encapsulation devices of beta cells for diabetes, which could be a great tool for exploring future islet transplantation strategies. The manuscript is quite well written with robust data to support the authors' concept, though the content may not be targeting wide range of readers.

Author response to Reviewer #1:

We thank the reviewer for his or her favorable overall review of the manuscript, and for his or her rigorous evaluation and identification of areas in need of further clarification and analysis. We believe that we can address all the reviewer's concerns. Most importantly, we have added significant clarifying text to the main body of the manuscript to clarify issues identified by the reviewer and performed supporting analyses. Our responses to each of the reviewer's questions are provided below.

Question 1. How is the hydrogel concentration considered for the in silico simulations, which critically affects the molecular diffusion? Authors used clinical-grade hydrogel but several products are available with different concentration and softness, which could be one of the essential variables to be included.

Response. The reviewer makes an observant point regarding the lack of clarification of the composition of the hydrogel considered in the simulations. Indeed, a variety of materials have been used for encapsulated islet delivery. As it was the intention of this work to explore the consequences of limited oxygen transport in generic devices, certain generalizations were necessary, including the selection of the simulated hydrogel material. We therefore considered this "generic" hydrogel to be 2% w/v alginate because of its common use in the field. In the previous writing, this detail was buried in page 9 of the Supplementary Information:

"A variety of hydrogels have been employed for islet encapsulation^{13,14}, thus it was necessary to select one representative material. Because of its

widespread use both presently and historically, we selected 2% (w/v) alginate as this model material.” (SI, p. 9)

To address the reviewer’s concern, we have added this point to the main text (in the **Results** section entitled “*SHARP enables evaluation of a diverse range of bioartificial pancreas devices*”). The modified passage is presented below for the reviewer’s convenience:

“In brief, oxygen from the host site, assumed to be at a constant 40 mmHg at the device-host boundary (Supplementary Fig. 9a), is transported in the hydrogel by passive diffusion. Herein, we are concerned with modeling transport in generic (i.e., representative) devices, thus it was necessary to select one material. Because of its widespread use both presently and historically (Strand et al., *Stem Cells Transl. Med.*, **6**, 1053–1058, 2017) we selected 2% (w/v) alginate as this model material, with oxygen permeability parameters obtained from the literature.” (p. 6)

The physical effect of varying hydrogel compositions, as the reviewer rightly indicates, is the molecular diffusivity. We would first like to note that the value we used for oxygen diffusivity in hydrogels, $2.7 \times 10^{-9} \text{ m}^2 \text{ s}^{-1}$, is a value near or exactly that used in similar modeling papers for both particular and generic hydrogel-based islet encapsulation devices (see Ch. 6 of Lewis, *Eliminating oxygen supply limitations for transplanted microencapsulated islets in the treatment of type 1 diabetes*, PhD dissertation, MIT, 2001; Dulong & Legallais, *Biotechnol. Bioeng.*, **96** 990–998, 2007; Buchwald et al., *BioMedical Engineering OnLine*, **14**, 28, 2015; and Buchwald et al., *Biotechnol. Bioeng.*, **115**, 232–245, 2018), and thus we were, in part, following the consensus of the most notable works in this field. As this discussion here, prompted by the reviewer’s excellent question, is quite relevant to the results of the paper, we have added these elaborations to the Supplementary Information (SI p. 9), where the selection of model parameters is discussed in greater detail.

In addition, it is worth noting that the value of the oxygen diffusion coefficient in hydrogel does indeed have a modest effect on model results. Over a reasonable range of values, 2×10^{-9} to $3 \times 10^{-9} \text{ m}^2 \text{ s}^{-1}$, i.e., between the diffusion coefficient values of oxygen in tissue and in water, the response of the mean $p\text{O}_2$ calculated in the islets in a test construct is nearly one-to-one proportional with change in the coefficient (**Fig. R1.1**). This is an obviously important analysis; thus, it has been added as **Supplementary Fig. 37** with relevant discussion added to the second paragraph of the **Discussion** section where SHARP’s limitations are discussed. Both are reproduced below:

Likewise, we considered all devices to use low concentration alginate hydrogel as the encapsulation matrix. However, a variety of hydrogel

materials and compositions have been explored for this purpose, which each may be endowed with oxygen permeabilities differing from the value used here. A sensitivity analysis of the diffusion coefficient of oxygen in hydrogel in a test construct suggests that a change in this parameter yields a roughly one-to-one change in the mean pO_2 of the islets over a reasonable range, thus, when modeling a specific device, accurate implementation of this parameter is indeed important (Supplementary Fig. 37). (p. 24).

Fig. R1.1 | (Supplementary Fig. 37) Model sensitivity to the diffusion coefficient of oxygen in hydrogel. a, b Schematic of the test construct (a) and mean pO_2 in the islets of the test construct versus the implemented value of the diffusion coefficient of oxygen in hydrogel ($D_{O_2,h}$) (b).

Again, we thank the reviewer for drawing attention to this essential point which certainly deserved more clarification in this manuscript.

Question 2. How did the authors integrate the O_2 -dependency of insulin-secreting ability of islets embedded in the BAP device? This may be the important factor for the success of BAP device in regard to the oxygenation strategy.

Response. The reviewer raises an important question regarding our implementation of oxygen-dependent maximum insulin secretion rate. This is obviously a critical part of almost all our analyses, and we regret leaving the mathematical description of this relationship to the supplementary information, where it is discussed in full (see SI p. 5 and **Supplementary Fig. 9c**). To address this, we have added that description to the main text **Methods** subsection, “Mass transport theory” (p. 27), adjusted for concision. It is reproduced below:

Oxygen-dependent insulin secretion potential (Supplementary Fig. 9c), S , was defined only in the cell clusters and modeled according to the Hill relationship (Buchwald et al., *Theor. Biol. Med. Model.*, **6**, 5, 2009; Buchwald et al., *Theor. Biol. Med. Model.*, **8**, 20, 2011; Buchwald et al., *Biotechnol. Bioeng.* **115**, 232–245, 2018), in favor of the bilinear (Suszynski et al., *J. Diabetes Res.*, **2016**, 7625947, 2016; Avgoustiniatos, PhD dissertation, MIT, 2001) or polynomial (Dulong & Legallais, *Biotechnol. Bioeng.*, **96**, 990–998, 2007) formulations used by others, based on studies of isolated islets exposed to variable media oxygen levels (Dionne et al., *Diabetes*, **42**, 12–21, 1993)

$$S = \begin{cases} 0, & p < p_N \\ \frac{p^{n_S}}{p^{n_S} + (K_S)^{n_S}}, & p \geq p_N \end{cases} \quad (9)$$

with half-maximal coefficient K_S (in mmHg) and Hill coefficient n_S (unitless). The step-down function again represents the loss of insulin secretion capacity of necrotic cells. More precisely, S defines the relative second-phase insulin secretion rate, rather than insulin concentration, and is hence not defined in the hydrogel. Accordingly, the volume-average insulin secretion potential of the cluster, \bar{S}_c , was given by

$$\bar{S}_c = \frac{1}{V_c} \iiint_C S \, dV \quad (10)$$

where C , the integral region, is the cell cluster and V_c is the volume of the cell cluster. We defined the loss of insulin secretion potential of the cell cluster, Ψ_c , simply as

$$\Psi_c = 1 - \bar{S}_c \quad (11)$$

and the loss of insulin secretion of the entire population of encapsulated islets is calculated as the weighted average of the individual islets, where the weights are each islet's volume. (p. 27)

To summarize for the reviewer, oxygen-dependent insulin secretion potential is modeled as a field existing within the cell clusters related to the pO_2 according to an essentially sigmoidal function, half-maximal at 2 mmHg (that is, if the local pO_2 is 2 mmHg, the insulin secretion potential is half of its maximum value at that point). The loss of insulin secretion potential is then given by integrating the values of that field over the domain of the cell cluster(s). This relationship was obtained by a prior research project which matched simulation results to empirical data collected by the Clark Colton group at MIT. We

appreciate the reviewer raising this issue and hope that our response has sufficiently clarified our implementation of this relationship in the model.

Question 3. In Fig.4, authors stated “An equivalent volume of islets from each fraction were then encapsulated in identical BAP devices.” Since the use of “equivalent” volume (300 IEQ) in the preparation between 2 groups of large fraction and small fraction is critically important that directly affects the transplantation outcome shown in Fig.4k, authors may demonstrate the data of absolute IEQ according to the size categories used for the in vivo transplantations.

Response. We appreciate the suggestion by the reviewer to demonstrate that the islet equivalent volume is equal in the large- and small-fraction devices by the traditional size categories and agree that this is a very critical point. The size group conversion table is produced below:

Table R1.1: Islet equivalence conversion chart for small- and large-fraction islets post partitioning.

Size group (µm)	K _{IEQ}	Small fraction		Large Fraction	
		Count	IEQ	Count	IEQ
0–50	0.005	304	1.52	96	0.48
51–100	0.167	1343	224.28	371	61.96
101–150	0.648	1053	682.34	265	171.72
151–200	1.685	373	628.51	213	358.91
201–250	3.500	46	161.00	155	542.50
251–300	6.315	0	0.00	58	366.27
301–350	10.352	0	0.00	9	93.17
>350	15.833	0	0.00	4	63.33
Net:			1697.65		1658.33

As indicated in the bolded net IEQ counts in the final row, the total islet volume was similar between the large- and small-fraction groups (~1658 versus ~1698 IEQ). To ensure that each of the devices contained precisely 300 IEQ, the large- and small-fraction islets were suspended in ~88 and ~91 µL of 2% w/v alginate (SLG100) to yield the same islet volumetric density, and finally ~16 µL of the alginate islet suspension was used for each device. This is now explained in the **Methods** section:

Using the conventional conversion table, the total rat islet yield obtained after partitioning was calculated at ~1658 and ~1698 IEQ for the large and

small fractions, respectively, to which ~88 and ~91 μ L of 2% w/v alginate was added such that the volumetric islet density was equivalent in both groups.” (p. 28)

We thank the reviewer for encouraging this analysis.

Question 4. Fig.4 demonstrated the animal studies using cylinder-shaped BAP system, to validate the SHARP in silico simulations. Since authors insist that the SHARP was developed that can be applied to variable types of BAP systems including planar slab and hollow cylinder (which authors later introduced the optimal configurations in Fig.5 for clinical settings), validity of SHARP may be tested not only in the single BAP system but in multiple systems (e.g. Cylinder shaped and Planar slab).

Response. The reviewer makes a helpful suggestion regarding testing SHARP’s predictions on other geometrical structures, however, there are several reasons why the bioartificial pancreas (BAP) device we tested in **Fig. 4** was uniquely suitable as a model BAP device in this project.

As a first point, the intention of this project is to derive general insights for the design of BAP devices. In the **Abstract**, we perhaps clumsily stated this aim through the phrase “a typical bioartificial pancreas”, which we have changed to “**generic** bioartificial pancreas **devices**” to better convey this intention. A similar change was made to the first paragraph of the **Discussion**. Second, the model BAP we selected is distinctive in that it can be almost completely described by the simple “generic” structure of an islet-containing hydrogel hollow cylinder with a passive core. Most other devices of the other shapes, which could in theory be used as alternative model devices, feature other nuances such as additional immunoprotective layers (Wang *et al.*, *Sci. Transl. Med.*, **13**, eabb4601, 2021) or vascularizing membranes (e.g., the PEC-Encap device by ViaCyte, San Diego, CA [Agulniuk *et al.*, *Stem Cells Transl. Med.*, **4**, 1–9, 2015]) which would need to be independently and rigorously accounted for, and would thus contradict our aim for this paper to serve as a general guide to BAP device design.

The reviewer may alternatively suggest to simply fabricate cylindrical or planar device comprised only of the islets and hydrogel themselves. However, as the reviewer pointed out in **Question 3**, obtaining equivalent islet volumes in the devices is critical for comparing graft performance, and the design of our model BAP enables that we can ensure this. As elaborated in the **Methods** section, the devices are fabricated by loading the modified thread in a mold (a polyethylene tube), and then filling the tube with a very small volume (~16 μ L) of 2% w/v alginate with suspended islets. Crosslinking then occurs by the radial diffusion of calcium ions from the surface of the thread core to the surrounding hydrogel matrix. When the device is eventually extruded from the mold, all

the alginate is crosslinked (containing the full islet load), and no residual alginate islet suspension is left within the mold. Additionally, even if we were able to fabricate a cylindrical device without the modified thread core, a purely hydrogel-based device also has the risk of fracturing after transplantation (see An *et al.*, PNAS, **115**, E263–E272, 2017, the source paper for the model BAP used herein, which shows that a cylindrical hydrogel was too brittle for transplant unless reinforced by the twisted suture thread). Thus, the BAP device we used was optimally suitable as a model for the system we wished to study.

We furthermore note that, because the islets were encapsulated at a low density and within a relatively thin structure, the results of **Fig. 4** were much more dependent on the islet size profiles than the geometrical shape of the BAP device. We can show this by calculating expected functional capacity and necrosis for islets from the small and large fractions encapsulated at the same volumetric density in, for example, a planar device of the same characteristic thickness (0.5 mm thickness, exposed to 40 mmHg oxygen only on one face):

Fig. R1.2 | a, b Net loss of function (**a**) and net necrosis (**b**) in the model BAP device used in the manuscript (the hollow cylinder) in and the analogous planar device of 0.5 mm thickness exposed to 40 mmHg only on one face. Statistics: two-way ANOVA with Sidak's post hoc p -value adjustment; **a** **** $p < 0.0001$ (all comparisons); **b** n.s. $p = 0.6757$, * $p = 0.0246$, *** $p < 0.0001$.

As is shown in **Fig. R1.2** above, though the absolute values of each functional outcome vary slightly (outcomes are logically slightly worse in the planar device for the same islet fraction), they are still quite similar, and more importantly, the magnitude of the difference between the devices containing large or small fraction islets is roughly the same, and therefore the overall directional conclusion is unchanged.

Despite our focus on generic devices in this project, we do indeed hope that SHARP will be used to guide the design of real devices in future applications. We thank the reviewer for this comment and hope that our response has addressed the sentiment behind his or her concern satisfactorily.

Question 5. I anticipate that the readers would have hard time to understand the heatmaps in Fig.5, although there are supplemental supporting materials (Suppl Fig.22). It would be great if authors could explain a bit more for better understanding in broad readers.

Response. We agree with the reviewer's assessment that the heatmaps in Fig. 5 may be difficult to understand and could benefit from more explanation. To address this concern, we made three changes. First, the introduction paragraph of the **Results** section in question was divided into two and adjusted to clarify the objective of the optimization study, as well as more precisely describe what the input variables were. The modified second paragraph of this section is reproduced below:

In this study, we considered a curative dose as the islet volume with the equivalent insulin secretion potential of 500 k IEQ fully functional islets (i.e., 500 k fIEQ), and varied the characteristic device thickness and cell density for three generic device shapes: the planar slab, the cylinder, and the hollow cylinder. The objective of the optimization was to minimize the characteristic curative device size under the constraint of some maximum amount of necrosis, which we call the necrotic tolerance (N_t). Because it is not well characterized how much cellular necrosis is permissible without inducing serious externalities, the optimal device, $S_{N_t}^$, was calculated under the constraint of three necrotic tolerances, volume fractions of 5%, 10%, and 20% of the total islet volume. In addition, we report the actual volume of islets, IEQ_{cure} , required to achieve the curative functional dose for the optimal devices. (p. 18)*

We believe that clarifying the language in this section should implicitly help make the heat maps more easily understood by the reader. Second, we added a sentence to the main text to explicitly describe the heat maps (reproduced below):

Heat maps were generated to illustrate the relationship between the inputs (τ and ρ , the horizontal and vertical axes, respectively) and the response variables (D_{cure} and N , shown as a colorimetric value for each combination of the inputs). (p. 18)

Note, the variables were defined in the previous sentence. Third, we modified language in the **Fig. 5** caption to better articulate the heat maps:

Heat maps (a) showing the curative device diameter (D_{cure} , left) and necrotic percentage (N , right), displayed as a colorimetric value represented by the top-positioned scale bar, for all combinations of the input parameters slab thicknesses (τ ; horizontal axis) and volumetric cell density (ρ ; vertical axis). (p. 20)

Descriptions of the heat maps for the cylindrical and hollow cylindrical structures were amended in the figure captions as well, albeit with less detail to avoid unnecessary repetition:

Heat maps (d) showing the curative device length (L_{cure} , left) and necrotic percentage (N , right) for all combinations of the input variables cylinder diameter (D) and cell density (ρ). ... Heat maps (g and j) showing the curative device length (L_{cure} , left) and necrotic percentage (N , right) for all combinations of the input variables hollow cylinder inner diameter (D_i) and cell density (ρ). (p. 20)

We hope that these changes cumulatively improved the clarity of the heat maps and of this analysis more generally. Again, we thank the reviewer for expressing concern over these plots as they are one of the focal points of this paper.

Concluding remarks.

The authors are thankful for this reviewer's attentive and detailed revision of the manuscript. The amendments to the text and supporting analyses performed on behalf of the reviewer's comments have undoubtedly improved the quality and rigor of this work. Finally, we hope that our responses to his or her individual comments have assuaged all his or her concerns.

Reviewer #2 (Remarks to the Author):

The manuscript describes a computational model that aims to predict bioartificial pancreas device performance by using randomly placed and sized islets to make the model more representative of real-life implementations. The model predicts a significant influence of endogenous variance in islet size distributions on device performance, and the manuscript also includes some limited in vitro and in vivo validation for this (e.g., a cell encapsulation device with pre-selected smaller islets performed better in diabetic mice than the same device with larger islets). Based on this, it also proposes a size conversion tool to calculate effective islet equivalents based on the specifics of the device used. The manuscript is well-written and organized, and it includes considerable amount of supporting information to document details of the work performed. The work as is, is mainly of interest for those in the bioengineering and artificial organ field focused on beta cell replacement. Major and minor issues that need to be addressed are summarized in detail below.

Author's response to Reviewer #2:

The quality of this reviewer's critique reveals his or her evident knowledge of the field of device-enabled islet delivery. We appreciate his or her thorough revision of the manuscript and for making several critiques and suggestions. In response to the reviewer, we have added several new references to relevant literature, reperformed the simulations of the model bioartificial pancreas devices to account for transport in the inert core, and revised the identified miscalculation. The amendments we have made to the manuscript on accord of the reviewer's comments have strengthened the rigor of this work. Our point-by-point responses to the reviewer's comments are provided below.

Major Comments

Question 1. The fact that smaller islets are likely to perform better than larger one when transplanted (directly or in a device) has been suggested by several other models before – some cited here (e.g., *Theor. Biol. Med. Model.* 2011, 8, 20), some not (e.g., *Artif. Organs* 2018, 42, E168). It is also why reaggregated islets of more uniform (and reduced) size are and have been explored for transplantation by several groups (e.g., *Diabetologia* 2010, 53, 937; *Tissue Eng Part A* 2013, 19, 604; *Diabetologia* 2018, 61, 2016). The novelty here is the implementation of the random distribution and its use to predict device-dependent islet equivalent conversion factors. Many of these considerations also hold for straightforward islet transplantation (i.e., not in a device) as those islet also rely on passive diffusion for several days and clustering is a likely reason for primary nonfunction in many cases. Maybe if this work could be extended to also incorporate such cases, e.g., at well vascularized sites such as the liver or kidney vs not well vascularized site such as

subcutaneous or fat pad, this might make it of more general interest and not just for the relatively small community of islet transplantation in bioartificial pancreas devices. Regarding this, it would probably be nice to also mention some of the publications suggesting that small islets perform better in transplantation than larger ones, e.g., Am. J. Physiol. Endocrinol. Metab. 2006, 290, E771; Diabetes 2007, 56, 594; Pancreas 2020, 49, 650.

Response. The reviewer rightly addresses that the evidence of the superior performance of smaller islets has been reported elsewhere. As a first point, we thank the reviewer for referring us to additional articles which have reported the benefits of smaller islet use or reaggregation. These references are now added to the introductory paragraph of the relevant **Results** section. We furthermore note that it was not our intention to claim the proposition of smaller islets performing better as a novel contribution of this work. Rather, we wished to communicate that this modeling methodology provides a way to calculate the magnitude of the estimated effect more precisely (as the reviewer implies), and given this possibility, to determine if endogenous variance in islet size distributions is sufficient to have a resulting effect on device performance, which we answer in the affirmative. We therefore also modified the introductory paragraph to this **Results** section, which is reproduced below for the reviewer's convenience:

As model predictions are sensitive to the assumed islet diameters, with smaller islets being favorable (Supplementary Figs. 10 and 11)(Buchwald et al., Biol. Med. Model., 8, 20, 2011; Iwata et al., Artif. Organs, 42, E168–E185, 2018), it may be extrapolated that the properties of the encapsulated islet size distribution are relevant for device success (Fig. 3). Indeed, the extent of diabetes correction has been correlated to islet size properties in non-encapsulated intraportal islet transplant recipients in the clinic (Suszynski et al., Transplantation, 97, 1286–1291, 2014) and smaller (MacGregor et al., Am. J. Physiol. Endocrinol. Metab., 290, E771–E779, 2006; Lehman et al., Diabetes, 56, 594–603, 2007; Komatsu et al., Pancreas, 49, 650–654, 2020) or reaggregated (and smaller)(O'Sullivan et al., Diabetologia, 53, 937–945, 2010; Ramachandran et al., Tissue Eng. Part A, 19, 604–612, 2013; Yu et al., Diabetologia, 61, 2016–2029, 2018) islets have been reported to yield better outcomes in preclinical investigations. Though it is logical to presume that smaller islets may have better outcomes due to oxygen constraints, it is not evident if the variance in isolated islet size profiles is sufficient to significantly influence BAP graft outcomes. Thus, using SHARP, we explored the degree of sensitivity of expected device performance to islet size distributions within the range of their natural expected variability (Fig. 3a). (p. 10)

We also appreciate the reviewer’s suggestion that considering direct islet transplantation in various sites would broaden the potentially interested readership of this work. While we agree with this assessment, we note that considering device-free implantation is unfortunately in many ways a different problem altogether than encapsulation, as each of the islets are in contact with the host tissue and may aggregate in certain regions. Because of this, we believe that a more sensible approach to model oxygen transport in directly transplanted islets would be to follow the method detailed in Suszynski *et al.*, *J. Diabetes Res.* **2016**, 7625947, 2016, a paper which considers intraportal islet transplantation. Therein, the authors model oxygen transport in an individual islet, and calculate the expected necrotic core volume for varying cases of islet diameter and boundary oxygen tension. We could theoretically expand this methodology herein and incorporate information about the islet size probability distributions to refine the calculations, however, we would then be moving beyond the core contributions of SHARP, which is to consider the spatial distributions of the islets as well. In short, we certainly agree that this is an interesting question deserving of rigorous attention, but it is beyond the intention of this work, which is to develop a “computational platform for optimizing the design of bioartificial pancreas devices”. Lastly, we note that consideration of different transplantation sites at variable boundary oxygen tensions is discussed in **Supplementary Fig. 36**, though of course only for encapsulated islets.

In sum, we thank the reviewer for emphasizing related research which should be referenced and for encouraging us to better articulate the novelty of our analysis in comparison to other approaches.

Question 2. Regarding fitting of the size distributions (e.g., Supplementary Fig. 3, Supplementary Table S4), for (theoretical) consistency it might be better to use either Weibull or log-normal across all fits despite differences in shapes. Ultimately, fit is quite similar, and I understand the authors’ desire to use the best fit, but it is hard to justify why would size distribution be a Weibull function for mouse but a lognormal for rat (and human) islets. I’d suggest picking one and using that uniformly across all, which would also allow more clear-cut comparisons, and just mention that the other one could be used as an alternate option with very similar result. On a related note, Supplementary Table 3 claims log normal as better fit based on RMSE. A quick recalculation with that data gave a much smaller difference for me (see below): RMSE of 0.0248 vs 0.0500 and not 0.0060 vs 0.0500 as shown there – please double check.

Size group	Observed frequency	Lognormal-predicted frequency	Weibull-predicted frequency
< 50	-	0.0662	0.0104

51–100	0.3319	0.3291	0.3339
101–150	0.4095	0.3993	0.4096
151–200	0.1537	0.1639	0.1645
201–250	0.0587	0.0207	0.0462
251–300	0.0253	0.0007	0.0117
301–350	0.0132	0.0000	0.0029
351–400	0.0078	0.0000	0.0008
	RMSE:	0.0500	0.0060 0.0248

Response. The reviewer makes a helpful suggestion to stick with one distribution type to describe the size probabilities of the islets/SC-βs. In fact, this was our original intention as well, for the same reasons the reviewer suggests, but after much thought we believe there are good reasons to use both distribution types, despite the sacrifice of simplicity. As a first point, we note that a theoretical justification has been elucidated which reveals both the lognormal and Weibull distributions as plausible functions to describe the size distributions of primary islets (Jo *et al.*, *Biophys J.*, **93**, 2655–2666, 2007). The shorthand description of this theory is that, during development, constituent islet cells may develop by coherent (wherein all cells of a representative cluster replicate or not at each discrete replication time) or independent (wherein each cell of a cluster replicates or not, independent of the other cells) dynamics, which ultimately result in cell clusters described by the two distributions. This is illustrated in the figure below, which is now added as **Supplementary Fig. 2** to the manuscript to clarify this for the reader:

Supplementary Fig. 2 Cell proliferation dynamics in primary islets result in size distributions described by lognormal or Weibull functions. Cell proliferation via coherent dynamics (all cells replicate together or not at each replication time) results in cell clusters whose size distribution is described by the lognormal function, whereas cell proliferation via independent dynamics (cells replicate or not, independent of each other, at each replication time) result in cell clusters whose size distribution is described by the Weibull function (Jo et al., *Biophys J.*, **93**, 2655–2666, 2007). Each circle represents a cell, and each horizontal row represents a discrete replication time.

In this paper, the authors note that it is likely that a combination of these proliferation dynamics may be present in the islets during development, and therefore the resulting size distributions may be more “lognormal-like” or “Weibull-like”. We thus view it as theoretically sound to select from the best fit distribution type.

There are also more practical problems with selecting one distribution type. For individual human islets, for example, whose distributions were obtained by the tabular method (rather than individual cluster traces), of the 129 human islet isolations analyzed, only 10 of them were found to be better described by the Weibull distribution, with many of the differences in best fits being quite significant. The sum of squared errors (SSE) for the lognormal and Weibull fits to the empirical data are presented below:

Fig. R2.1 | Lognormal and Weibull fits to individual human islet size frequency data. **** $p < .0001$, paired student's t -test.

Thus, given the theoretical backing and ability to refine curve fits using both functions, we are electing to keep both equations to describe the size distributions. To nevertheless address the spirit of the reviewer's comment (that is, to allow for more clear-cut comparisons between cell sources), we amended **Supplementary Table 4**, where the size distribution parameters are tabulated, to include values for both distribution types. It is reproduced on the following page:

Supplementary Table 4 Islet and SC- β size distribution properties. Best-fit distribution parameters (α and β) and mean diameter (d) on a number and volume basis (subscripts n and v , respectively) determined for each cell source. Mean diameters provided only next to the distribution type determined to be the best fit, according to the RMSE.

Cell source	Best-fit distribution parameters				\bar{d}_n (μm)	\bar{d}_v (μm)
	α_n	β_n	α_v	β_v		
Best-fit lognormal parameters						
Rat islets [‡]	0.40	112.6	0.40	182.0	122.0	197.1
Juvenile porcine islets [‡]	0.32	152.2	0.32	207.2	179.4	218.1
Human islets [‡]	0.36	115.4	0.36	170.2	123.2	181.6
Mouse islets	0.30	149.7	0.30	196.1		
NN SC- β s	0.14	149.5	0.14	158.6		
MM SC- β s	0.14	241.7	0.14	256.3		
Best-fit Weibull parameters						
Rat islets	2.78	131.5	3.48	228.4		
Juvenile porcine islets	3.53	193.4	4.18	254.8		
Human islets	2.89	126.3	2.39	242.0		
Mouse islets [‡]	3.67	168.5	5.00	206.3	152.0	189.4
NN SC- β s [‡]	7.87	157.6	9.29	166.3	148.3	157.7
MM SC- β s [‡]	8.14	254.7	9.58	267.3	240.0	253.8

[†]Probability density functions of the lognormal and Weibull distributions are given by Eqs. (S1) and (S2); cumulative probability density functions are given by Eqs. (S3) and (S4).

[‡]Symbol indicates best-fit distribution type for that species, defined by lower RMSE.

Regarding the discrepancy discovered in **Supplementary Table 3**, the authors are impressed by the reviewer's attention to detail and grateful for his or her uncovering of this error. Rechecking the Excel file where this was calculated, it seems that a human error was made when transcribing the table in Word (the SSE was calculated at 0.0006 so it is our guess that this is the source of the mistake). However, and while it does not make a difference in the best-fit conclusion, we should nonetheless note that both our calculation and the reviewer's calculation is actually the square root of the SSE, rather than the RMSE, which also accounts for the number of observations, n :

$$\text{RMSE} = \sqrt{\frac{\sum_i^n (x_i - \hat{x}_i)^2}{n}} \quad (\text{Eq. 2.1})$$

where x_i is the observed value and \hat{x}_i is the estimated (or predicted) value. Therefore, RMSE values for the Weibull- and lognormal-predicted frequencies are now listed to correct that error, at values of 0.0189 and 0.0094, respectively. We thank the reviewer for evaluating our selection of the best-fit curves and for identifying a (fortunately inconsequential) mistake in our presented calculations.

Question 3. Effect of fibrosis on worsening hypoxia is not mentioned at all except that fibrotic cell coverage appeared to be more substantial for the device with larger islets (Supplementary Fig. 20). This could also significantly alter device performance after a few days, and the estimated IEQ calculations could be overestimates for devices likely to be exposed to more severe fibrotic coverage. One possible reason why the LF-BAP performed worse in vivo than predicted by the model. Also, exhaustion of remaining cells due to increasing metabolic demand as functional mass is being reduced over time is not included in the model – another possible reason for overpredicting performance.

Response. We agree with the reviewer that both fibrosis and metabolic exhaustion are probable reasons which explain SHARP's overprediction for the functional islet equivalence in the LF-BAPs. However, there is some evidence that fibrosis may not be the only explicating factor for this observation. Of the four LF-BAPs, two devices showed some degree of maintained cell survival (LF-BAP 2 and LF-BAP 4; see **Supplementary Fig. 19**), but these devices also experienced a higher degree of fibrotic coverage. In fact, islet survival in the LF-BAP with the least amount of fibrosis (LF-BAP 1) was perhaps worst of all judging from the histological samples. The fibrosis quantification is reproduced below with annotations connecting each data point to the samples (this update was included in **Supplementary Fig. 21d**):

Fig. R2.2 (Supplementary Fig. 21d) | Fibrosis quantification in the retrieved LF-BAPs and SF-BAPs, with the device number indicated offset to the right. Note, there is no relation between LF-BAP 1 and SF-BAP 1, the numbers refer to the device order listed in histology figures **Supplementary Figs. 19 and 20**. *n.s.* ($p = 0.0844$); unpaired two-sided student's *t*-test.

Thus, at the very least, the magnitude of fibrotic deposition was not correlated to islet survival in this experiment. But to communicate the general sentiment of this comment to the readers, we updated the concluding paragraph of that results section to explicitly state these points. Relevant parts of the paragraph are reproduced below, with changes highlighted in yellow:

The prediction that device function is significantly dependent on the islet size distribution, with smaller, less polydisperse ones being favored was generally supported by the results of this study. With respect to expected functional capacity, the magnitude of the difference was perhaps underpredicted by SHARP, given the nearly complete loss of function in some LF-BAPs. One possibility is that the greater fibrotic coverage in most LF-BAPs suffocated the grafts. Additionally, because islet OCR is correlated to local glucose conditions, SHARP indicated that the survival and function of the encapsulated rat islets would be reduced if exposed to high glucose, which may be expected in the hyperglycemic LF-BAP recipients. We speculate that, in addition to well documented deleterious effects of metabolic exhaustion resulting from chronic hyperglycemia as functional cell mass is gradually lost over time (Ottosson-Laakso, et al., Diabetes, 66, 3013–3028, 2017), elevated BG levels due to poor graft

function may induce sustained increases in islet OCR, in turn worsening islet function and survival and may thus accelerate graft attrition in the mouse model (Supplementary Fig. 21). (p. 15)

We thank the reviewer for drawing our attention to this point and hope that our amendments to the text and supporting analysis are satisfactory.

Question 4. For the modeling (e.g., mass transport) it is claimed that “All BAP systems modeled consisted of two subdomains, the hydrogel matrix (denoted by subscript h) and the encapsulated cell clusters (denoted by subscript c). What about devices with inert core, such as the nylon suture – what oxygen transport is assumed there? In Supplementary Fig. 16c,d, it looks like there is oxygen transport in the core nylon suture - is that true? I don't think that is realistic, maybe for the covering PMMA layer only.

Response. We thank the reviewer for addressing this issue. It is indeed prudent to consider oxygen transport (or the lack thereof) in the central nylon suture thread, and in the original analysis, it was simply assumed that transport in this region matched that of oxygen in hydrogel, which is obviously not the case. While oxygen transport is likely still present in the nylon suture, it is significantly slower than in hydrogel: the reported diffusion coefficient of oxygen in nylon is $5.00 \times 10^{-13} \text{ m}^2 \text{ s}^{-1}$ (Kjeldsen, Water Research, **27**, 121–131, 1993), i.e., four orders of magnitude lower than that in aqueous media. Thus, we reperformed the simulations of the model BAPs containing islets from the large and small fractions, treating the internal scaffold as a homogeneous nylon material. The schematics for the simulation are provided below (and are likewise presented in the updated **Supplementary Fig. 17b**):

Fig. R2.3 (Supplementary Fig. 17b) | Schematics (left) and illustrated dimensions (right) of the BAP simulated using SHARP.

Note, the two dark grey internal spheres at the end represent knots in the twisted nylon suture that are required to maintain the torsion in the thread as a product of its fabrication process (An *et al.*, PNAS, **115**, E263–E272, 2017). A representative image of one device is provided below:

Fig. R2.4 | Representative image of the knot in the twisted suture.

While the consideration of the knots likely only exerts a small influence, we felt that it was nonetheless worthy of inclusion to address the general spirit of the reviewer’s comment, which is to model the device as accurately as possible.

The results of the updated simulation are nonetheless quite similar to those of the previous model, and most importantly do not change the statistical conclusions. Results from the previous simulation are compared to the new results in the figure below (corresponding to **Fig. 4f–h**):

Fig. R2.5 (Fig. 4f–h) | Net loss of function (left), expected IIEQ (middle), and net necrosis (right) calculated by SHARP originally (top row) and after implementing treatment of the scaffold as an independent material.

Text in the **Results** section was also updated to reflect the new values associated with this analysis, as well as the description of the “Mass transport theory” section of the **Methods**. Results in the supplementary information (**Supplementary Fig. 17**) were also updated with the new simulation data. The whole supplemental figure is reproduced on the following page with the updated caption, with textual changes highlighted in yellow:

Supplementary Fig. 17 Rat islet-containing BAPs. **a** Representative stereo microscope images of the model cell encapsulation device used for testing the influence of islet size distributions on diabetes corrective capacity; the image on the right shows a representative image of the knot at the end of the twisted suture used to maintain torsion in the thread, integrated in the model as a sphere near the end of the thread. **b**

*Schematics (of the LF-BAP and SF-BAP, left and right, respectively) and dimensions used in SHARP to simulate the model BAP. **c, d** Surface plots showing the spatial pO_2 (p) distribution in a center cut plane of one iteration of the LF-BAP (**c**) and SF-BAP (**d**) simulations. White space **within simulated islets** indicates regions of expected necrosis. **e, f** SHARP-predicted necrotic volume percentage, N_c , versus islet diameter, d , of individual islets in the LF-BAPs ($n = 801$) (**e**) and in the SF-BAPs ($n = 2,576$) (**f**); data shown were collected from 4 iterations of each device containing an islet volume of 300 IEQ (aggregated data shown in Fig. 4h).*

As the selection of the material of the inert core was found to have a minimal impact on model predictions, we did not rerun simulations for the “optimization study” with the hollow core geometry results (**Fig. 5g–l**), which were in any case performed to analyze “generic” devices. We thank the reviewer for encouraging us to model the BAP device more rigorously in the work pertaining to **Fig. 4** and hope that our adjustments have assuaged the reviewer’s concerns in full.

Question 5. The prediction of function implemented for adjusted IEQ conversion coefficients (e.g., Fig. 6) seems to be independent of species; if it is, why? Oxygen consumption rates are significantly higher in rodents (e.g., mice) than in humans (Table S5), so loss of function should be higher too - why is this not coming up in f_{IEQ} ?

Response. The reviewer asks an important question regarding the species dependency of our definition of functional islet equivalence. First, we note that the data in **Fig. 6b** compares predictions from the finite element model (SHARP) versus the machine learning surrogate (SHARP-ML) in two types of devices, both of which contained human islets. In fact, the machine learning surrogate was developed exclusively to model behavior of human islet containing devices as an initial proof-of-concept, because of the wealth of data generated from the optimization study (**Fig. 5**), and for its greater clinical significance. We speculate that the confusion for rat islets may be a result of our use of a similar hue of red in **Fig. 6** as that which we used to denote rat islets in all other figures, so as a first point, we changed the figure color scheme, and more clearly annotated **Fig. 6b** to indicate that the devices both contained human islets:

Fig. R2.6 | Former (left) and updated (right) **Fig. 6**, featuring an updated color scheme and new annotations to the device description images in the table caption of **b** to mitigate cell source confusion.

Similar color changes were also made in the relevant supplementary figures (**Supplementary Figs. 33 and 34**).

Thus, we agree with the reviewer that incorporation of rat islets in the same devices would yield higher loss of function and thus lower functional islet equivalence, because of its higher OCR. We can briefly confirm this by comparing the functional islet equivalence conversion coefficients and net loss of function in **Fig. 6b**'s "Hypothetical device 1" with the same dimensions (a slab of 0.55 mm thickness) and cell density ($\rho = 2.5\%$ v/v) containing 500 IEQ of human and rat islets:

Fig. R2.7 | Functional islet equivalence in identical devices containing human or rat islets. **a** Conventional islet equivalence conversion coefficients (κ_{IEQ}) versus functional islet equivalence conversion coefficients (κ_{fIEQ}) for a representative device (slab with a thickness of 0.55 mm and cell density of 2.5% v/v) containing 500 IEQ of either human or rat islets. **b** Net fIEQ in the representative device containing human or rat islets (dashed line indicates 500 IEQ, the simulated net islet volume). **** $p < 0.0001$ (two-sided student's t -test with Welch's correction). Data was collected from SHARP with 20 simulations of the representative device containing 500 IEQ islets.

As is shown in **Fig. R2.7** above, the device of equivalent properties containing rat islets shows lower functional equivalence of the larger islet size groups and for the islet population in aggregate. We thank the reviewer for identifying this source of confusion and hope that our responses have clarified this issue for the reader.

Minor Comments

1. N/A

Concluding remarks.

The authors are very grateful for this knowledgeable reviewer's thorough and detailed critique of our work. His or her comments inspired us to refine the simulations of the model BAP devices (in **Fig. 4**) and improve the clarity and impact of the manuscript in many areas. We hope that our amendments to the manuscript and supporting analyses in our responses to the reviewer's individual comments have addressed all his or her concerns.

Reviewer #3 (Remarks to the Author):

The ML application is really interesting, since the computational time in this kind of application (several hours) is not acceptable. Moreover, the ML approach seems to improve the performances in the case of bigger size. Only few comments below to be consider before publication:

Author Response to Reviewer #3:

We thank the reviewer for his concise and overall favorable review of the manuscript. Following the reviewer's suggestions, we have added significant elaborations to improve the clarity of the surrogate machine learning model. Our responses to the reviewer's individual comments are included below.

Question 1. The authors stated that they used a "neural network" (NN) model: several different types of NNs are available, which one has been used here? Also, the parameters used for the other techniques should be reported in the supplementary material. In general, details to understand the ML methodologies are not reported and should be added.

Response. We thank the reviewer for requesting specification of the selection of the neural network model used herein and for his or her suggestion to report the details of the ML methodologies. To answer the reviewer here, the neural network was a multi-layer perceptron with a rectified linear activation function (ReLU) and the following variables: (1) number of hidden layers and (2) hidden units in each layer that were selected during the hyperparameter tuning process. This information has been added to the **Methods** subsection "Development of SHARP-ML", the entire paragraph of which is reproduced below with changes indicated by yellow highlights:

Six machine-learning models were selected as constituents of an ensemble to provide a range of functional approaches (i.e., model structures) to learn the associations between κ_{fIEQ} values and the predictor variables (Fig. 6a). These included LightGBM⁵⁶, XGboost⁵⁷, Cubist⁵⁸ rules, kernel K-nearest neighbors⁵⁹, a simple linear model, and a neural network model. The neural network model selected was a multi-layer perceptron with a rectified linear activation function (ReLU) with the following variables: (1) number of hidden layers and (2) hidden units in each layer that were selected during the hyperparameter tuning process. Prior to model training, the training data was subset for each geometry, and the hyperparameters for each model were tuned for each subset. The training data consisted of the κ_{fIEQ} values

generated using SHARP in the optimization study (Fig. 5). For each of the constituent models, the Pearson correlation coefficient was estimated for different subgroups of the data (defined for each geometry in Supplementary Fig. 32) and were used as weights for each constituent model's contribution to the final ensemble. More information is provided in Supplementary Materials in subsections Methods Machine Learning Model Development and Supplementary Data 1–5. (p. 28)

Regarding the details to understand the ML methodology, we agree that it deserves a more detailed description. Thus, we have included the specifications for each constituent ML model and geometry in an extensive additional document, Supplementary Data File 1. This file reports the R functions used for hyperparameter tuning and post-tuning model specifications for each geometry and constituent ML model. A representative image of Supplementary Data File 1 is provided below:

Hyper parameter tuning function

The hyperparameters for each model were tuned using a hyperparameter tuning workflow set created by the R function below.

```
## build workflow set of models
## @param model_rec recipe: preprocessing steps
build_workflow_set <- function(model_rec){

  # linear model spec
  lm_spec <-
    linear_reg() %>%
    set_engine('lm') %>%
    set_mode('regression')

  # lightGBM spec
  gbm_spec <-
    boost_tree(mtry = tune(),
               trees = tune(),
               min_n = tune(),
               tree_depth = tune(),
               loss_reduction = tune(),
               learn_rate = tune(),
               sample_size = 0.75) %>%
    set_engine("lightgbm") %>%
    set_mode("regression")

  # xgboost spec
  xgb_spec <-
    boost_tree(mtry = tune(),
               trees = tune(),
               min_n = tune(),
               tree_depth = tune(),
               loss_reduction = tune(),
```

Fig. R3.1 | Representative screenshot of Supplementary Data File 1, added to report the details of the ML model development.

We also note that the code may be accessed by the reviewer (and the readers) via a Github link (<https://github.com/scworland/sharp-ml>) reported in the Code Availability section of the article. Again, we appreciate the reviewer for encouraging us to improve the description of SHARP-ML's development.

Question 2. The results are related to SHARP-ML approach without defining in detail which approach between the 6 proposed is the optimal one and why.

Response. We appreciate the reviewer for posing this question. Defining which model was the optimal one unfortunately does not have a straightforward answer, as different models were better suited for different regions of the parameter space. **Supplementary Figure 32** (produced on the following page of this document) reports the prediction quality for each constituent ML model for subgroups of the input data. For example, when looking at the hollow cylinder geometry, the extreme gradient boosting (XGB) model performs generally well except for large diameters, large islet densities, and large islet size groups, whereas either the linear model (lm) or the cubist model would be preferred. It is for this reason that the correlation coefficients were used as weights for a weighted ensemble average of each model for each combination of input parameters, as we state in the main text, to “*leverage the unique strengths of each model for different subgroups of the parameter space*” (p. 21).

Because this was not clear to the reviewer, who is clearly an expert in machine learning, we have addressed this shortcoming in the updated manuscript in two ways. First, we added explicit text to the **Methods** subsection “Development of SHARP-ML” which clarifies the “ensemble” nature of the ML approach: “*For each of the underlying machine-learning models, the Pearson correlation coefficients were estimated for different subgroups of the data (defined for each geometry in Supplementary Fig. 32) and were used as weights for each constituent model’s contribution to the final ensemble*” (p. 28). Second, we updated **Supplementary Fig. 32**, which presents the absolute values of the underlying machine learning model weights for defined data subgroups, with a guide to aid readers in understanding the presented data. It is reproduced on the following page with the updated caption aimed at clarification:

Supplementary Fig. 32 Constituent machine learning model weights. a Guide for interpreting model weight charts in **b**: performance of the constituent machine learning models, defined by the Pearson’s correlation coefficient (r) is shown in four major subgroups of the parameter space, for each islet size group. The Pearson’s r is used as the weights of each underlying model’s contribution to the ensemble in the regions of the parameter space. **b** Absolute weights (Pearson’s r) for each machine learning model, geometry, and combinations of cell density (ρ) and geometry variable (cylinder diameter, D , planar slab thickness, τ , or hollow cylinder inner diameter, D_i) grouped by their relationship to the median values for each geometry.

Concluding remarks.

We are grateful for the reviewer's rigorous attention to the machine learning modeling component of this project. The additional material we have added in response to the reviewer's general concern regarding the dearth of explanation of the machine learning methodologies have certainly improved the manuscript. We hope that these additional contributions and our responses to the individual questions have addressed his or her concerns in full.

REVIEWERS' COMMENTS

Reviewer #1 (Remarks to the Author):

Authors addressed the reviewers comments with reasonable responses and revised manuscript.

Reviewer #2 (Remarks to the Author):

The authors have done considerable revisions to the manuscript to address the reviewers' comments. While they chose to not follow several of the suggestions, they at least provided detailed justifications for doing so and incorporated some changes to address or discuss the highlighted deficiencies. Overall, some drawbacks notwithstanding, this is an impressive amount of work with a rigorous modeling approach. The authors' commitment is also well illustrated by the fact that the answer to the reviewers' comments is on its own more than 30 pages long with detailed comments for each issue raised. As already noted, the work as is, is mainly of interest for the relatively narrow bioengineering and artificial organ field focused on beta cell replacement, but I do not have any further comments regarding the content of the manuscript.

Reviewer #3 (Remarks to the Author):

The authors addressed all my comments satisfactorily.